# Augmentation Component Analysis: Modeling Similarity via the Augmentation Overlaps

**Lu Han, Han-Jia Ye✉, De-Chuan Zhan**
State Key Laboratory for Novel Software Technology, Nanjing University
{hanlu,yehj}@lamda.nju.edu.cn, zhandc@nju.edu.cn

## Abstract

Self-supervised learning aims to learn a embedding space where semantically similar samples are close. Contrastive learning methods pull views of samples together and push different samples away, which utilizes semantic invariance of augmentation but ignores the relationship between samples. To better exploit the power of augmentation, we observe that semantically similar samples are more likely to have similar augmented views. Therefore, we can take the augmented views as a special description of a sample. In this paper, we model such a description as the augmentation distribution, and we call it augmentation feature. The similarity in augmentation feature reflects how much the views of two samples overlap and is related to their semantical similarity. Without computational burdens to explicitly estimate values of the augmentation feature, we propose Augmentation Component Analysis (ACA) with a contrastive-like loss to learn principal components and an on-the-fly projection loss to embed data. ACA equals an efficient dimension reduction by PCA and extracts low-dimensional embeddings, theoretically preserving the similarity of augmentation distribution between samples. Empirical results show that our method can achieve competitive results against various traditional contrastive learning methods on different benchmarks. Code available at https://github.com/hanlu-nju/AugCA.

## 1 Introduction

The rapid development of contrastive learning has pushed self-supervised representation learning to unprecedented success. Many contrastive learning methods surpass traditional pretext-based methods by a large margin and even outperform representation learned by supervised learning (Wu et al., 2018; van den Oord et al., 2018; Tian et al., 2020a; He et al., 2020; Chen et al., 2020a;c). The key idea of self-supervised contrastive learning is to construct views of samples via modern data augmentations (Chen et al., 2020a). Then discriminative embeddings are learned by pulling together views of the same sample in the embedding space while pushing apart views of others.

Contrastive learning methods utilize the semantic invariance between views of the same sample, but the semantic relationship between samples is ignored. Instead of measuring the similarity between certain augmented views of samples, we claim that the similarity between the *augmentation distributions* of samples can reveal the sample-wise similarity better. In other words, semantically similar samples have similar sets of views. As shown in Figure 1 left, two images of deer create many similar crops, and sets of their augmentation results, *i.e.*, their distributions, overlap much. In contrast, a car image will rarely be augmented to the same crop as a deer, and their augmentation distributions overlap little. In Figure 1 right, we verify the motivation numerically. We approximate the overlaps between image augmentations with a classical image matching algorithm (Zitova & Flusser, 2003), which counts the portion of the key points matched in the raw images. We find samples of the same class overlap more than different classes on average, supporting our motivation. Therefore, we establish the semantic relationship between samples in an unsupervised manner based on the similarity of augmentation distributions, *i.e.*, how much they overlap.

In this paper, we propose to describe data directly by their augmentation distributions. We call the feature of this kind the *augmentation feature*. The elements of the augmentation feature represent

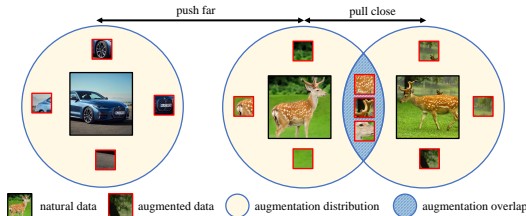 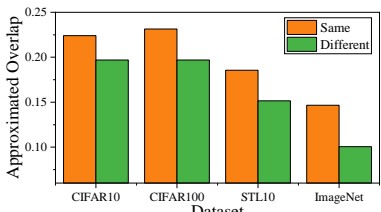

Figure 1: Left: semantically similar samples (*e.g.*, those in the same class) usually create similar augmentations. The right figure indicates the same class images have higher averaged augmentation overlaps than those from different classes on four common datasets. For this reason, we learn embeddings by preserving the similarity between augmentation distributions of samples.

the probability of getting a certain view by augmenting the sample as shown in the left of Figure 2. The augmentation feature serves as an "ideal" representation since it encodes the augmentation information without any loss and we can easily obtain the overlap of two samples from it. However, not only its elements are hard to calculate, but also such high-dimensional embeddings are impractical to use.

Inspired by the classical strategy to deal with high-dimensional data, we propose Augmentation Component Analysis (ACA), which employs the idea of PCA (Hotelling, 1933) to perform dimension reduction on augmentation features previously mentioned. ACA reformulates the steps of extracting principal components of the augmentation features with a contrastive-like loss. With the learned principal components, another on-the-fly loss embeds samples effectively. ACA learns operable low-dimensional embeddings theoretically preserving the augmentation distribution distances.

In addition, the similarity between the objectives of ACA and traditional contrastive loss may explain why contrastive learning can learn semantic-related embeddings – they embed samples into spaces that partially preserve augmentation distributions. Experiments on synthetic and real-world datasets demonstrate that our ACA achieves competitive results against various traditional contrastive learning methods. Our contributions are as follows:

- We propose a new self-supervised strategy, which measures sample-wise similarity via the similarity of augmentation distributions. This new aspect facilitates learning embeddings.
- We propose ACA method that implicitly employs the dimension reduction over the augmentation feature, and the learned embeddings preserve augmentation similarity between samples.
- Benefiting from the resemblance to contrastive loss, our ACA helps explain the functionality of contrastive learning and why they can learn semantically meaningful embeddings.

## 2 RELATED WORK

**Self-Supervised Learning.** Learning effective visual representations without human supervision is a long-standing problem. Self-supervised learning methods solve this problem by creating supervision from the data itself instead of human labelers. The model needs to solve a pretext task before it is used for the downstream tasks. For example, in computer vision, the pretext tasks include colorizing grayscale images (Zhang et al., 2016), inpainting images (Pathak et al., 2016), predicting relative patch (Doersch et al., 2015), solving jigsaw puzzles (Noroozi & Favaro, 2016), predicting rotations (Gidaris et al., 2018) and exploiting generative models (Goodfellow et al., 2014; Kingma & Welling, 2014; Donahue & Simonyan, 2019). Self-supervised learning also achieves great success in natural language processing (Mikolov et al., 2013; Devlin et al., 2019).

**Contrastive Learning and Non-Contrastive Methods.** Contrastive approaches have been one of the most prominent representation learning strategies in self-supervised learning. Similar to the metric learning in supervised scenarios (Ye et al., 2019; 2020), these approaches maximize the agreement between positive pairs and minimize the agreement between negative pairs. Positive pairs are commonly constructed by co-occurrence (van den Oord et al., 2018; Tian et al., 2020a; Bachman et al., 2019) or augmentation of the same sample (He et al., 2020; Chen et al., 2020a;c; Li et al., 2021; Ye et al., 2023), while all the other samples are taken as negatives. Most of these methods employ the InfoNCE loss (van den Oord et al., 2018), which acts as a lower bound of mutual information between

views. Based on this idea, there are several methods that attempt to improve contrastive learning, including mining nearest neighbour (Dwibedi et al., 2021; **?**; Azabou et al., 2021) and creating extra views by mixing up (Kalantidis et al., 2020) or adversarial training (Hu et al., 2021). Another stream of methods employs a similar idea of contrastive learning to pull views of a sample together without using negative samples (Grill et al., 2020; Chen & He, 2021). Barlow Twins (Zbontar et al., 2021) minimizes the redundancy within the representation vector. Tsai et al. (2021) reveals the relationship among Barlow Twins, contrastive and non-contrastive methods. Most of these methods only utilize the semantic invariance of augmentation and ignore the relationship between samples. Different from them, we propose a new way to perform self-supervised learning by preserving the similarity of augmentation distribution, based on the observation that a strong correlation exists between the similarity of *augmentation distributions* and the similarity of semantics.

**Explanation of Contrastive Learning.** Several works provide empirical or theoretical results for explaining the behavior of contrastive learning. Tian et al. (2020b); Xiao et al. (2021) explore the role of augmentation and show contrastive model can extract useful information from views but also can be affected by nuisance information. Zhao et al. (2021) empirically shows that contrastive learning preserves low-level or middle-level instance information. In theoretical studies, Saunshi et al. (2019) provide guarantees of downstream linear classification tasks under conditionally independence assumption. Other works weaken the assumption but are still unrealistic (Lee et al., 2021; Tosh et al., 2021). HaoChen et al. (2021) focus on how views of different samples are connected by the augmentation process and provide guarantees with certain connectivity assumptions. Wang et al. (2022) notice that the augmentation overlap provides a ladder for gradually learning class-separated representations. In addition to the alignment and uniformity as shown by Wang & Isola (2020), Huang et al. (2021) develop theories on the crucial effect of data augmentation on the generalization of contrastive learning. Hu et al. (2022) explain that the contrastive loss is implicitly doing SNE with "positive" pairs constructed from data augmentation. Inspired by the important role of augmentation, we provide a novel self-supervised method that ensures preserving augmentation overlap.

## 3 NOTATIONS

The set of all natural data (data without augmentation) is denoted by $\bar{\mathcal{X}}$, with size $|\bar{\mathcal{X}}| = N$. We assume that the natural data follow a uniform distribution $p(\bar{x})$ on $\bar{\mathcal{X}}$, *i.e.*, $p(\bar{x}) = \frac{1}{N}, \forall \bar{x} \in \bar{\mathcal{X}}$. By applying an augmentation method $\mathcal{A}$, a natural sample $\bar{x} \in \bar{\mathcal{X}}$ could be augmented to another sample $x$ with probability $p_{\mathcal{A}}(x \mid \bar{x})$, so we use $p(\cdot \mid \bar{x})$ to encode the augmentation distribution. [1] For example, if $\bar{x}$ is an image, then $\mathcal{A}$ can be common augmentations like Gaussian blur, color distortion and random cropping (Chen et al., 2020a). Denote the set of all possible augmented data as $\mathcal{X}$. We assume $\mathcal{X}$ has finite size $|\mathcal{X}| = L$ and $L > N$ for ease of exposition. Note that $N$ and $L$ are finite, but can be arbitrarily large. We denote the encoder as $f_{\theta}$, parameterized by $\theta$, which projects a sample $x$ to an embedding vector in $\mathbb{R}^k$.

## 4 LEARNING VIA AUGMENTATION OVERLAPS

As we mentioned in Section 1, measuring the similarity between the augmentation distributions, *i.e.*, the overlap of the augmented results of the two samples reveals their *semantic* relationship well. For example, in natural language processing, we usually generate augmented sentences by dropping out some words. Then different sentences with similar meanings are likely to contain the same set of words and thus have a high probability of creating similar augmented data. With the help of this self-supervision, we formulate the embedding learning task to meet the following similarity preserving condition:

$$\mathrm{d}_{\mathbb{R}^k} \left( f_{\theta^\star}\left(\bar{x}_1\right), \ f_{\theta^\star}\left(\bar{x}_2\right)\right) \propto \mathrm{d}_{\mathcal{A}}(p(\cdot \mid \bar{x}_1), \ p(\cdot \mid \bar{x}_2)) \,. \tag{1}$$

$\mathrm{d}_{\mathbb{R}^k}$ is a distance measure in the embedding space $\mathbb{R}^k$, and $\mathrm{d}_{\mathcal{A}}$ measures the distance between two augmentation distributions. Equation (1) requires the learned embedding with the optimal parameter $\theta^\star$ has the *same similarity comparison* with that measured by the augmentation distributions. In this section, we first introduce the *augmentation feature* for each sample, *which is a manually designed embedding satisfying the condition in Equation* (1). To handle the high dimensionality and complexity of the augmentation feature, we further propose our Augmentation Component Analysis (ACA) that learns to reduce the dimensionality and preserve the similarity.

---

[1] Note that $p(\cdot \mid \bar{x})$ is usually difficult to compute and we can only sample from it. We omit the subscript $\mathcal{A}$ and directly use $p(\cdot \mid \bar{x})$ in the following content for convenient

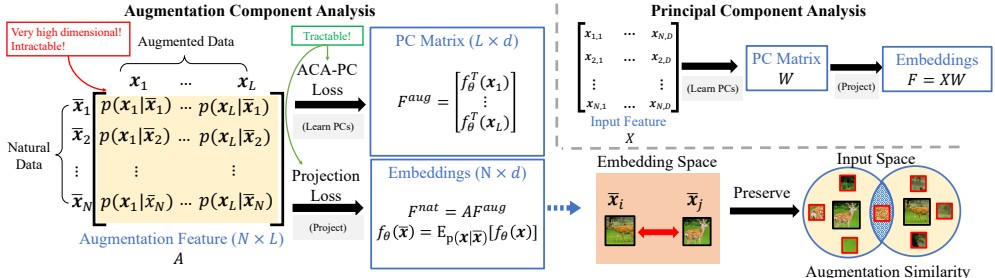

Figure 2: The idea of learning embeddings via Augmentation Component Analysis (ACA). The upper right figure demonstrates the process of PCA. It learns PCs and projects the input feature to get embeddings of data. Similarly, ACA performs PCA on the augmentation feature, which encodes all the information about the augmentation distribution. To overcome the dimensional and computational complexity, ACA employs ACA-PC loss and projection loss to learn PCs and embeddings tractably. Via ACA, our model can learn embeddings that preserve augmentation similarity for natural data.

## 4.1 AUGMENTATION FEATURE

To reach the goal of similarity preserving in Equation (1), a direct way is to manually construct the feature by the augmentation distributions of each natural sample, *i.e.*, $f(\bar{\boldsymbol{x}}) = [p(\boldsymbol{x}_1 \mid \bar{\boldsymbol{x}}), \ldots, p(\boldsymbol{x}_L \mid \bar{\boldsymbol{x}})]^\top$, where each element $p(\boldsymbol{x}_i \mid \bar{\boldsymbol{x}})$ represents the probability of getting a certain element $\boldsymbol{x}_i$ in space $\mathcal{X}$ by augmenting $\bar{\boldsymbol{x}}$. We omit $\theta$ in $f(\bar{\boldsymbol{x}})$ since such *augmentation feature*[2] does not rely on any learnable parameters. In this case, any distance $\mathrm{d}_{\mathbb{R}^L}$ defined in the space of $f$ is exactly a valid distribution distance, which reveals the augmentation overlaps and is related to the semantic similarity.

Although the constructive augmentation feature naturally satisfies the similarity preserving condition (Equation (1)) (because it directly use the augmentation distribution without loss of information), it is impractical for the following reasons. First, its dimensionality is exponentially high, which is up to $L$, the number of possible augmented results. For example, even on CIFAR10, the small-scale dataset with image size $32 \times 32 \times 3$, $L$ is up to $256^{3072}$ (3072 pixels and 256 possible pixel values). Second, the computation of each element is intractable. We may need an exponentially large number of samples to accurately estimate each $p(\boldsymbol{x} \mid \bar{\boldsymbol{x}})$. The dimensionality and computation problems make the augmentation feature impractical both at inference and training time. Such inconvenience motivates us to (1) conduct certain dimension reduction to preserve the information in low dimensional space (Section 4.2) and (2) develop an efficient algorithm for dimension reduction (Section 4.3).

## 4.2 DIMENSION REDUCTION ON AUGMENTATION FEATURES

To deal with the high-dimensional property, we employ the idea of PCA (Hotelling, 1933), which reconstructs the data with principal components.[3] For convenience, we denote the design matrix of augmentation feature by $A$, where $A \in \mathbb{R}^{N \times L}$, $A_{\bar{\boldsymbol{x}}, \boldsymbol{x}} = p(\boldsymbol{x} \mid \bar{\boldsymbol{x}})$ (see Figure 2). We perform PCA on a transformed augmentation feature called *normalized augmentation feature*:

$$\hat{A} = AD^{-\frac{1}{2}}, \tag{2}$$

where $D = \mathrm{diag}([d_{\boldsymbol{x}_1}, d_{\boldsymbol{x}_2}, \ldots, d_{\boldsymbol{x}_L}])$, $d_{\boldsymbol{x}} = \sum_{\bar{\boldsymbol{x}}} p(\boldsymbol{x} \mid \bar{\boldsymbol{x}})$. Based on normalized augmentation feature, we can develop an efficient algorithm for similarity preserving embeddings.

Assume the SVD of $\hat{A} = U\Sigma V^\top$ with $U \in \mathbb{R}^{N \times N}$, $\Sigma \in \mathbb{R}^{N \times L}$, $V \in \mathbb{R}^{L \times L}$, PCA first learns the projection matrix consisting of the top-$k$ right singular vectors, which can be denoted as $\tilde{V} \in \mathbb{R}^{L \times k}$. The vectors in $\tilde{V}$ are called Principal Components (PCs). Then, it projects the feature by $\hat{A}\tilde{V}$ to get the embeddings for each sample. The overall procedure is illustrated at the top-right of Figure 2. But performing PCA on the augmentation feature will encounter many obstacles. The element of augmentation feature is not possible to estimate accurately, not to mention its high dimensionality.

---

[2]Following the common knowledge in dimension reduction, we call the raw high dimensional representation as "feature", and learned low-dimensional representation as "embedding".

[3]In this paper, we use the non-centred version (Reyment & Jvreskog, 1996), which is more appropriate for observations than for variables, where the origin matters more.

Even if we can somehow get the projection matrix $\tilde{V}$, it is also impractical to project the high-dimensional matrix $\hat{A}$. For this reason, *we propose ACA to make PC learning and projection process efficient without explicitly calculating elements of augmentation feature.*

### 4.3 AUGMENTATION COMPONENT ANALYSIS

Although there are several obstacles when performing PCA on the augmentation features directly, fortunately, it is efficient to sample from the augmentation distribution $p(\boldsymbol{x} \mid \bar{\boldsymbol{x}})$, *i.e.*, by performing augmentation on the natural data $\bar{\boldsymbol{x}}$ and get an augmented sample $\boldsymbol{x}$. Being aware of this, our ACA uses two practical losses to simulate the PCA process efficiently by sampling. The first contrastive-like loss leads the encoder to learn principal components of $\hat{A}$, which can be efficiently optimized by sampling like traditional contrastive methods. The second loss performs on-the-fly projection of $\hat{A}$ through the training trajectory, which solves the difficulty of high dimensional projection.

**Learning principal components.** ACA learns the principal components by an efficient contrastive-like loss. Besides its projection functionality, these learned principal components can also serve as embeddings that preserve a kind of posterior distribution similarity, as we will show later.

In the SVD view, $U\Sigma$ serves as the PCA projection results for samples and $V$ contains the principal components (Jolliffe, 2002). However, if changing our view, $V\Sigma$ can be seen as the representation of each column. Since each column of $\hat{A}$ encodes the probability of the augmented data given natural data, $V\Sigma$ preserves certain augmentation relationships, as we will show in Theorem 4.2 later. To leverage the extrapolation power of encoders like deep neural networks, we choose to design a loss that can guide the parameterized encoder $f_\theta$ to learn similar embeddings as PCA. Inspired by the rank minimization view of PCA (Vidal et al., 2016), we employ the low-rank approximation objective with matrix factorization, similar to HaoChen et al. (2021):

$$\min_{F \in \mathbb{R}^{L \times k}} \mathcal{L}_{mf} = \|\hat{A}^\top \hat{A} - FF^\top\|_F^2 \,, \tag{3}$$

where columns of $F$ store the scaled version of top-$k$ right singular vectors, and each row can be seen as the embedding of augmented data as will show in Lemma 4.1. According to Eckart–Young–Mirsky theorem (Eckart & Young, 1936), by optimizing $\mathcal{L}_{mf}$, we can get the optimal $\hat{F}$, which has the form $\tilde{V}\tilde{\Sigma}Q$, $Q \in \mathbb{R}^{k \times k}$ is an orthonormal matrix. $\tilde{\Sigma}$ and $\tilde{V}$ contains the top-$k$ singular values and right singular vectors. By expanding Equation (3), we get Augmentation Component Analysis Loss for learning Principal Components (ACA-PC) in the following lemma:

**Lemma 4.1** (ACA-PC loss). *Let $F_{\boldsymbol{x},:} = \sqrt{d_{\boldsymbol{x}}} f_\theta^\top(\boldsymbol{x}), \forall \boldsymbol{x} \in \mathcal{X}$. Minimizing $\mathcal{L}_{mf}$ is equivalent to minimizing the following objective:*

$$\begin{aligned}
\mathcal{L}_{ACA\text{-}PC} = &- 2\mathbb{E}_{\substack{\bar{\boldsymbol{x}} \sim p(\bar{\boldsymbol{x}}), \, \boldsymbol{x}_i \sim p(\boldsymbol{x}_i|\bar{\boldsymbol{x}}) \\ \boldsymbol{x}_j \sim p(\boldsymbol{x}_j|\bar{\boldsymbol{x}})}} f_\theta(\boldsymbol{x}_i)^\top f_\theta(\boldsymbol{x}_j) \\
&+ N\mathbb{E}_{\boldsymbol{x}_1 \sim p_\mathcal{A}(\boldsymbol{x}_1), \boldsymbol{x}_2 \sim p_\mathcal{A}(\boldsymbol{x}_2)} \left[ \left( f_\theta(\boldsymbol{x}_1)^\top f_\theta(\boldsymbol{x}_2) \right)^2 \right].
\end{aligned} \tag{4}$$

The proof can be found in Appendix F. In ACA-PC, the first term is the common alignment loss for augmented data and the second term is a form of uniformity loss (Wang & Isola, 2020). Both terms can be estimated by Monte-Carlo sampling. ACA-PC is a kind of contrastive loss. But unlike most of the others, it has theoretical meanings. We note that the form of ACA-PC differs from spectral loss (HaoChen et al., 2021) by adding a constant $N$ before the uniformity term. This term is similar to the noise strength in NCE (Gutmann & Hyvärinen, 2010) or the number of negative samples in InfoNCE (van den Oord et al., 2018). It can be proved that the learned embeddings by ACA-PC preserve the posterior distribution distances between augmented data:

**Theorem 4.2** (Almost isometry for posterior distances). *Assume $f_\theta$ is a universal encoder, $\sigma_{k+1}$ is the $(k+1)$-th largest singular value of $\hat{A}$, $d_{\min} = \min_{\boldsymbol{x}} d_{\boldsymbol{x}}$, and $\delta_{\boldsymbol{x}_1\boldsymbol{x}_2} = \mathbb{I}(\boldsymbol{x}_1 = \boldsymbol{x}_2)$, the minimizer $\theta^*$ of $\mathcal{L}_{ACA-PC}$ satisfies:*

$$\mathrm{d}_{\mathrm{post}}^2(\boldsymbol{x}_1, \boldsymbol{x}_2) - \frac{2\sigma_{k+1}^2}{d_{\min}} \left(1 - \delta_{\boldsymbol{x}_1\boldsymbol{x}_2}\right) \leq \|f_{\theta^*}(\boldsymbol{x}_1) - f_{\theta^*}(\boldsymbol{x}_2)\|_2^2 \leq \mathrm{d}_{\mathrm{post}}^2(\boldsymbol{x}_1, \boldsymbol{x}_2) \,, \quad \forall \boldsymbol{x}_1, \boldsymbol{x}_2 \in \mathcal{X}$$

*where the posterior distance*

$$\mathrm{d}_{\mathrm{post}}^2(\boldsymbol{x}_1, \boldsymbol{x}_2) = \sum_{\bar{\boldsymbol{x}} \in \bar{\mathcal{X}}} \left( p_\mathcal{A}(\bar{\boldsymbol{x}} \mid \boldsymbol{x}_1) - p_\mathcal{A}(\bar{\boldsymbol{x}} \mid \boldsymbol{x}_2) \right)^2 \tag{5}$$

measures the squared Euclidean distance between the posterior distribution $p_{\mathcal{A}}(\bar{\boldsymbol{x}} \mid \boldsymbol{x}) = \frac{p(\boldsymbol{x}|\bar{\boldsymbol{x}})p(\bar{\boldsymbol{x}})}{p_{\mathcal{A}}(\boldsymbol{x})}$.

We give the proof in Appendix G. Theorem 4.2 states that the optimal encoder for ACA-PC preserves the distance of posterior distributions between augmented data within an error related to embedding size $k$. As $k$ increase to $N$, the error decrease to $0$. It corresponds to the phenomenon that a larger embedding size leads to better contrastive performance (Chen et al., 2020a). The posterior distribution $p_{\mathcal{A}}(\bar{\boldsymbol{x}} \mid \boldsymbol{x})$ represents the probability that a given augmented sample $\boldsymbol{x}$ is created by a natural sample $\bar{\boldsymbol{x}}$. Augmented data that are only produced by the same natural sample will have the smallest distance, and embeddings of those in overlapped areas will be pulled together by ACA-PC. Since the overlapped area are usually created by two same-class samples, ACA-PC can form semantically meaningful embedding space.

It is also noticeable that the optimal encoder meets the similarity preserving condition (Equation (1)) but concerning the posterior distribution for augmented data not the augmentation distribution for natural data. Since what we care about is the distribution of natural data, we further propose a projection loss that helps learn good embeddings for all the natural data.

**On-the-fly Projection.** As stated in the previous part, the learned embeddings by ACA-PC not only serve as certain embeddings for augmented data but also contain principal components of normalized augmentation feature. Based on this, we propose to use these embeddings to act as a projection operator to ensure meaningful embeddings for all the natural data. To be specific, denote the embedding matrix for all augmented data as $F^{aug}(\in \mathbb{R}^{L \times k})$, where each row $F^{aug}_{\boldsymbol{x},:} = f^{\top}_{\theta^*}(\boldsymbol{x})$. From Equation (3) and $\hat{F}_{\boldsymbol{x},:} = \sqrt{d_{\boldsymbol{x}}} f^{\top}_{\theta^*}(\boldsymbol{x})$, it can be easily seen that:

$$F^{aug} = D^{-\frac{1}{2}}\hat{F} = D^{-\frac{1}{2}}\tilde{V}\tilde{\Sigma}Q$$

Similar to PCA (Hotelling, 1933) that projects the original feature by the principal components $V$, we propose to use $F^{aug}$ to project the augmentation feature to get the embeddings for each natural sample. Denote the embedding matrix for natural data as $F^{nat}(\in \mathbb{R}^{N \times k})$, where each row $F^{nat}_{\bar{\boldsymbol{x}},:}$ represents the embeddings of $\bar{\boldsymbol{x}}$. We compute $F^{nat}$ as follows:

$$F^{nat} = A F^{aug} = \hat{A} D^{\frac{1}{2}} D^{-\frac{1}{2}} \tilde{V}\tilde{\Sigma}Q = (\tilde{U}\tilde{\Sigma})\tilde{\Sigma}Q, \tag{6}$$

where $\tilde{\Sigma}, \tilde{U}$ contain the top-$k$ singular values and corresponding left singular vectors. It is noticeable that $F^{nat}$ is exactly the PCA projection result multiplied by an additional matrix $\tilde{\Sigma}Q$. Fortunately, such additional linear transformation does not affect the linear probe performance (HaoChen et al., 2021). With Equation (6), the embedding of each natural sample can be computed as follows:

$$F^{nat}_{\bar{\boldsymbol{x}},:} = A_{\bar{\boldsymbol{x}},:} F^{aug} = \sum_{\boldsymbol{x}} p(\boldsymbol{x} \mid \bar{\boldsymbol{x}}) f^{\top}_{\theta^*}(\boldsymbol{x}) = \mathbb{E}_{\boldsymbol{x} \sim p(\boldsymbol{x}|\bar{\boldsymbol{x}})} f^{\top}_{\theta^*}(\boldsymbol{x}) \tag{7}$$

which is exactly the expected feature over the augmentation distribution. Similar to Theorem 4.2, the embeddings calculated by Equation (7) also present a certain isometry property:

**Theorem 4.3** (Almost isometry for weighted augmentation distances). *Assume $f_\theta$ is a universal encoder, $\sigma_{k+1}$ is the $(k+1)$-th largest sigular value of $\hat{A}, \delta_{\bar{\boldsymbol{x}}_1 \bar{\boldsymbol{x}}_2} = \mathbb{I}(\bar{\boldsymbol{x}}_1 = \bar{\boldsymbol{x}}_2)$, let the minimizer of $\mathcal{L}_{ACA-PC}$ be $\theta^*$ and $g(\bar{\boldsymbol{x}}) = \mathbb{E}_{\boldsymbol{x} \sim p(\boldsymbol{x}|\bar{\boldsymbol{x}})} f_{\theta^*}(\boldsymbol{x})$ as in Equation (7), then:*

$$\mathrm{d}^2_{\text{w-aug}}(\bar{\boldsymbol{x}}_1, \bar{\boldsymbol{x}}_2) - 2\sigma^2_{k+1}(1 - \delta_{\bar{\boldsymbol{x}}_1 \bar{\boldsymbol{x}}_2}) \leq \|g(\bar{\boldsymbol{x}}_1) - g(\bar{\boldsymbol{x}}_2)\|^2_{\Sigma_k^{-2}} \leq \mathrm{d}^2_{\text{w-aug}}(\bar{\boldsymbol{x}}_1, \bar{\boldsymbol{x}}_2), \quad \forall \boldsymbol{x}_1, \boldsymbol{x}_2 \in \mathcal{X}$$

*where $\|\cdot\|_{\Sigma_k^{-2}}$ represent the Mahalanobis distance with matrix $\Sigma_k^{-2}, \Sigma_k = \mathrm{diag}([\sigma_1, \sigma_2, \dots, \sigma_k])$ is the diagonal matrix containing top-$k$ singular values and the weighted augmentation distance*

$$\mathrm{d}^2_{\text{w-aug}}(\bar{\boldsymbol{x}}_1, \bar{\boldsymbol{x}}_2) = \frac{1}{N} \sum_{\boldsymbol{x} \in \mathcal{X}} \frac{(p(\boldsymbol{x} \mid \bar{\boldsymbol{x}}_1) - p(\boldsymbol{x} \mid \bar{\boldsymbol{x}}_2))^2}{p_{\mathcal{A}}(\boldsymbol{x})} \tag{8}$$

*measures the weighted squared Euclidean distance between the augmentation distribution $p(\boldsymbol{x} \mid \bar{\boldsymbol{x}})$.*

Different from Theorem 4.2, which presents isometry between Euclidean distances in embeddings and augmentation distribution, Theorem 4.3 presents isometry between Mahalanobis distances. The weighted augmentation distances weigh the Euclidean distances by $p_{\mathcal{A}}(\boldsymbol{x})$. $\mathrm{d}_{\text{w-aug}}$ can be regarded as a valid augmentation distance measure $\mathrm{d}_{\mathcal{A}}$ as in Equation (1) and $F^{nat}$ preserve such a distance. So our goal is to make embeddings of $\bar{\boldsymbol{x}}$ approaches $\mathbb{E}_{p(\boldsymbol{x}|\bar{\boldsymbol{x}})} f_{\theta^*}(\boldsymbol{x})$. However, as stated before, the additional projection process is not efficient, i.e., we need exponentially many samples from $p(\boldsymbol{x} \mid \bar{\boldsymbol{x}})$. We notice that samples during the training process of ACA-PC can be reused. For this reason, we

propose an on-the-fly projection loss that directly uses the current encoder for projection:

$$\mathcal{L}_{\text{proj}} = \mathbb{E}_{\bar{\boldsymbol{x}} \sim p(\bar{\boldsymbol{x}})} \left[ \| f_\theta(\bar{\boldsymbol{x}}) - \mathbb{E}_{p(\boldsymbol{x}|\bar{\boldsymbol{x}})} f_\theta(\boldsymbol{x}) \|_2^2 \right] \tag{9}$$

**Full objective of ACA.** Based on the discussion of the above parts, ACA simultaneously learns the principal components by ACA-PC and projects natural data by an on-the-fly projection loss. The full objective of ACA has the following form:

$$\mathcal{L}_{\text{ACA-Full}} = \mathcal{L}_{\text{ACA-PC}} + \alpha \mathcal{L}_{\text{proj}} \tag{10}$$

where $\alpha$ is a trade-off hyperparameter. We also find $N$ in Equation (4) too large for stable training, so we replace it with a tunable hyperparameter $K$. Here, we only display the loss in expectation forms. The details of the implementation are described in Appendix A.

## 5    A PILOT STUDY

In this section, we experiment with our Augmentation Component Analysis method on a synthetic mixture component data with a Gaussian augmentation method. In this example, we aim to show the relationship between semantic similarity and posterior/weighted augmentation distances. We also show the effectiveness of our method compared to traditional contrastive learning. In this example, the natural data $\bar{\boldsymbol{x}}$ are sampled from a mixture gaussian with $c$ component:

$$p(\bar{\boldsymbol{x}}) = \sum_{i=1}^{c} \pi_i \mathcal{N}(\boldsymbol{\mu}_i, s_i I)$$

We use Gaussian noise as the data augmentation of a natural data sample, *i.e.*, $\mathcal{A}(\bar{\boldsymbol{x}}) = \bar{\boldsymbol{x}} + \xi$ where $\xi \sim \mathcal{N}(0, s_a I)$. Concretely, we conduct our experiment on 2-D data with $c = 4, \pi_i = \frac{1}{c}, s_i = 1$ and $\boldsymbol{\mu}_i$ uniformly distributed on a circle with radius 2 . For each component, we sample 200 natural data with the index of the component as their label. For each natural datum, we augment it 2 times with $s_a = 4$, which results in totally 1600 augmented data. We compute the augmentation probability for between $\boldsymbol{x}$ and $\bar{\boldsymbol{x}}$ by $p(\boldsymbol{x} \mid \bar{\boldsymbol{x}})$ and we normalize the probability for each $\bar{\boldsymbol{x}}$.

First, we plot the distribution of posterior distances (Equation (5)) for pairs of augmented data and weighted augmentation distances (Equation (8)) for pairs of natural data in Figure 3 left. The two distances appear to have similar distributions because the synthetic data are Gaussian. It can be seen that data from the same component tend to have small distances, while from different components, their distances are large. In low-distance areas, there are pairs of the same class, which means that the two distances are reliable metrics for judging semantic similarity. In all, this picture reveals the correlation between semantic similarity and posterior/weighted augmentation distances.

Second, we compare our methods with SimCLR (Chen et al., 2020a), the traditional contrastive method and Spectral (HaoChen et al., 2021), which similarly learns embeddings with spectral theory. We test the learned embeddings using a Logistic Regression classifier and report the error rate of the prediction in Figure 3 right. We also report performance when directly using *augmentation feature* (AF). First, AF has discriminability for simple linear classifiers. SimCLR and Spectral tend to underperform AF as the embedding size increases, while our methods consistently outperform. It may be confusing since our method performs dimension reduction on this feature. But we note that as the embedding size increases, the complexity of the linear model also increases, which affects the generalizability. All the methods in  Figure 3 right show degradation of this kind. However, our methods consistently outperform others, which shows the superiority of ACA. Additionally, by adding projection loss, ACA-Full improves ACA-PC by a margin. Additionally, traditional contrastive learning like SimCLR achieves similar performance as our methods. We think it reveals that traditional contrastive learning has the same functionality as our methods.

## 6    EXPERIMENTS

### 6.1    SETUP

**Dataset.** In this paper, we conduct experiments mainly on the following datasets with RTX-3090 $\times 4$. **CIFAR-10** and **CIFAR-100** (Krizhevsky et al., 2009): two datasets containing totally 500K images of size $32 \times 32$ from 10 and 100 classes respectively. **STL-10** (Coates et al., 2011): derived from ImageNet (Deng et al., 2009), with $96 \times 96$ resolution images with 5K labeled training data from 10 classes. Additionally, 100K unlabeled images are used for unsupervised learning. **Tiny**

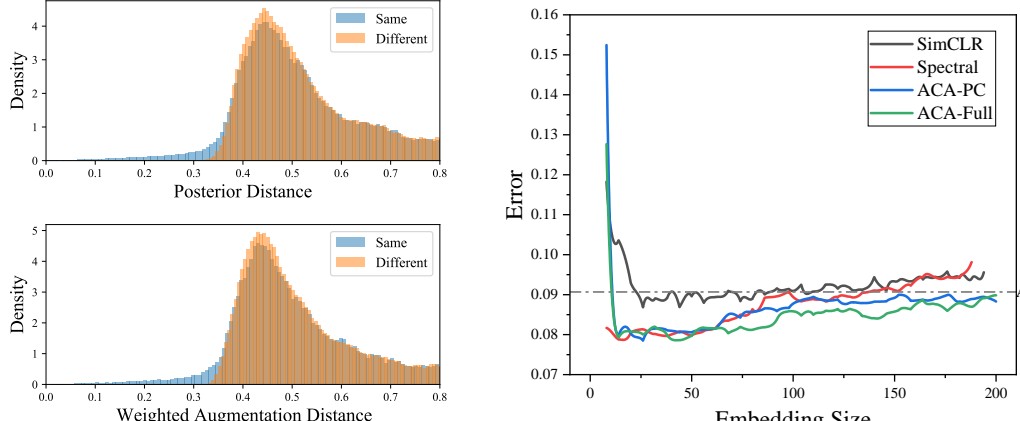

Figure 3: Synthetic experiments on mixture Gaussian data with Gaussian noise as augmentation. (a) The posterior distance and weighted augmentation distances among data sampled from the same component and different components. It reveals the correlation between semantic similarity and the two distances, especially when the distance is small. (b) Comparison of linear classification performance among SimCLR, Spectral and our methods with various embedding dimensions ranging from 4 to 200. The dashed line represents the result when directly using *Augmentation Feature* (AF). ACA-PC outperforms SimCLR and Spectral. ACA-Full further improves.

**ImageNet**: a reduced version of ImageNet (Deng et al., 2009), composed of 100K images scaled down to $64 \times 64$ from 200 classes. **ImageNet-100** (Tian et al., 2020a): a subset of ImageNet, with 100-classes. **ImageNet** (Deng et al., 2009), the large-scale dataset with 1K classes.

**Network Structure.** Following common practice (Chen et al., 2020a;b;c), we use the encoder-projector structure during training, where the projector projects the embeddings into a low-dimensional space. For CIFAR-10 and CIFAR-100, we use the CIFAR variant of ResNet-18 (He et al., 2016; Chen & He, 2021) as the encoder. We use a two-layer MLP as the projector whose hidden dimension is half of the input dimension and output dimension is 64. For STL-10 and Tiny ImageNet, only the max-pooling layer is disabled following (Chen & He, 2021; Ermolov et al., 2021). For these two datasets, we use the same projector structure, except that the output dimension is 128. For ImageNet, we use ResNet-50 with the same projector as Chen et al. (2020a).

**Image Transformation.** Following the common practice of contrastive learning (Chen et al., 2020a), we apply the following augmentations sequentially during training: (a) crops with a random size; (b) random horizontal flipping; (c) color jittering; (d) grayscaling. For ImageNet-100 and ImageNet, we use the same implementation as (Chen et al., 2020a).

**Optimizer and other Hyper-parameters.** For datasets except for ImageNet, adam optimizer (Kingma & Ba, 2015) is used for all datasets. For CIFAR-10 and CIFAR-100, we use 800 epochs with a learning rate of $3 \times 10^{-3}$. For Tiny ImageNet and STL-10, we train 1,000 epochs with a learning rate $2 \times 10^{-3}$. We use a 0.1 learning rate decay at 100, 50, 20 epochs before the end. Due to hardware resource restrictions, we use a mini-batch of size $512$. The weight decay is $1 \times 10^{-6}$ if not specified. Following common practice in contrastive learning, we normalize the projected feature into a sphere. For CIFAR-10, we use $\alpha = 1$. For the rest datasets, we use $\alpha = 0.2$. By default, $K$ is set to 2. For ImageNet, we use the same hyperparameters as (Chen et al., 2020a) except batch size being 256, $\alpha = 0.2$ and $K = 2$.

**Evaluation Protocol.** We evaluate the learned representation on two most commonly used protocols – linear classification (Zhang et al., 2016; Kolesnikov et al., 2019) and k-nearest neighbors classifier (Chen & He, 2021). In all the experiments, we train the linear classifier for 100 epochs. The learning rate exponentially decays from $10^{-2}$ to $10^{-6}$. The weight decay is $1 \times 10^{-6}$. We report the classification accuracy on test embeddings as well as the accuracy of a 5-Nearest Neighbors classifier for datasets except for ImageNet.

## 6.2 PERFORMANCE COMPARISON

In Table 1, we compare the linear probe performance on various small-scale or mid-scale benchmarks with several methods including SimCLR (Chen et al., 2020a), BYOL (Grill et al., 2020),

Table 1: Top-1 linear classification accuracy and 5-NN accuracy on four datasets with a ResNet-18 encoder. We use **bold** to mark the best results and underline to mark the second-best results. $\sharp$ means the results are reproduced by our code.

| method | CIFAR-10 | | CIFAR-100 | | STL-10 | | Tiny ImageNet | |
|---|---|---|---|---|---|---|---|---|
| | Linear | 5-NN | Linear | 5-NN | Linear | 5-NN | Linear | 5-NN |
| SimCLR$^\sharp$ | 90.88 | 88.25 | 65.53 | 55.32 | 89.27 | 85.44 | 47.41 | 31.95 |
| BYOL$^\sharp$ | 91.20 | 89.52 | 64.85 | 54.92 | 88.64 | **86.73** | 47.66 | 32.96 |
| SimSiam$^\sharp$ | 91.06 | 89.43 | 65.41 | 54.84 | 90.04 | 85.48 | 45.17 | 30.41 |
| Spectral$^\sharp$ | 90.28 | 87.25 | 65.42 | 55.05 | 89.16 | 84.23 | 45.69 | 29.32 |
| ACA-PC (ours) | 90.35 | 87.38 | 65.69 | 54.57 | 90.08 | 85.86 | 46.08 | 30.97 |
| ACA-Full (ours) | **92.04** | **89.79** | **67.16** | **56.52** | **90.88** | 86.44 | **48.79** | **33.53** |

Table 2: Left: Top-1 classification accuracy and 5-NN accuracy on ImageNet-100 with ResNet-18. [†]: results are taken from (Wang & Isola, 2020). [⋆]: results are taken from (Tian et al., 2020b).$^\sharp$ means the results are reproduced by our code. Right: Top-1 classification accuracy on ImageNet with ResNet-50, results are taken from (Chen & He, 2021; HaoChen et al., 2021). We use **bold** to mark the best results and underline to mark the second-best results.

| ImageNet-100 | Linear | 5-NN |
|---|---|---|
| MoCo[†] | 72.80 | - |
| $\mathcal{L}_{align} + \mathcal{L}_{uniform}$ [†] | 74.60 | - |
| InfoMin[⋆] | 74.90 | - |
| SimCLR$^\sharp$ | 75.62 | 62.70 |
| Spectral$^\sharp$ | 75.52 | 61.80 |
| ACA-PC (ours) | 75.80 | 62.54 |
| ACA-Full (ours) | **76.02** | **63.20** |

| ImageNet | Linear (100 epochs) |
|---|---|
| SimCLR | 66.5 |
| MoCo v2 | 67.4 |
| BYOL | 66.5 |
| SimSiam | 68.1 |
| Spectral | 66.97 |
| ACA-PC (ours) | 67.21 |
| ACA-Full (ours) | **68.32** |

SimSiam (Chen & He, 2021) and Spectral (HaoChen et al., 2021). For transfer learning benchmarks, please refer to Appendix D and Appendix E. SimCLR uses is a method that uses contrastive loss. BYOL and SimSiam do not use negative samples. Spectral is a similar loss derived from the idea of spectral clustering. From Table 1, we can see that our ACA-Full method achieves competitive results on small- or mid-scale benchmarks, achieving either the best or the second-best results on all benchmarks except the 5-NN evaluation on STL-10. Also, ACA-PC differs from ACA-Full in the projection loss. In all the benchmarks, we can see that the projection loss improves performance.

For large-scale benchmarks, we compare several methods on ImageNet-100 and ImageNet. On ImageNet-100, we compare our method additionally to MoCo (He et al., 2020), $\mathcal{L}_{align} + \mathcal{L}_{uniform}$ (Wang & Isola, 2020) and InfoMin (Tian et al., 2020b). Note that the results of the other three methods are reported when using the ResNet-50 encoder, which has more capacity than ResNet18. Our method can also achieve state-of-the-art results among them. This means that our method is also effective with relatively small encoders even on large-scale datasets. On ImageNet, we see that ACA-PC achieves competitive performance against state-of-the-art contrastive methods (Chen et al., 2020a;c; Grill et al., 2020; Chen & He, 2021; HaoChen et al., 2021) and ACA-Full achieves the best.

## 7 Conclusion and Future Work

In this paper, we provide a new way of constructing self-supervised contrastive learning tasks by modeling similarity through augmentation overlap, which is motivated by the observation that semantically similar data usually creates similar augmentations. We propose Augmentation Component Analysis to perform PCA on augmentation feature efficiently. Interestingly, our methods have a similar form as the traditional contrastive loss and may explain the ability of contrastive loss. We hope our paper can inspire more thoughts about how to measure similarity in self-supervised learning and how to construct contrastive learning tasks. Future studies may be explorations of applying ACA to learn representations of other forms of instances, such as tasks (Achille et al., 2019) and models (Wu et al., 2023).

## ACKNOWLEDGE

This research was supported by NSFC (61773198, 62006112,61921006), Collaborative Innovation Center of Novel Software Technology and Industrialization, NSF of Jiangsu Province (BK20200313)

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

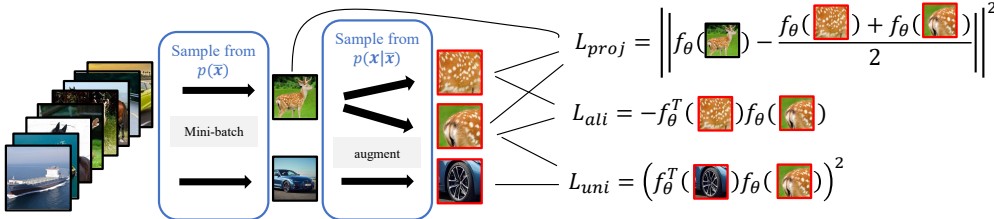

Figure 4: The implementation of ACA. Like traditional contrastive learning methods, ACA samples a mini-batch from the whole natural dataset, then performs two random augmentations on the mini-batch. The mini-batch sampling and augmentation create samples from $p(\bar{x})$ and $p(x \mid \bar{x})$ respectively. Then we use the samples to estimate the values of $\mathcal{L}_{\text{uni}}, \mathcal{L}_{\text{uni}}$ and $\mathcal{L}_{\text{proj}}$ as in the figure.

## A    IMPLEMENTATION AND FURTHER DISCUSSION

In the previous section, we have presented the expected form of ACA. Thanks to its form, we can efficiently optimize ACA by Monte-Carlo sampling, making the problem **tractable**.

For convenience of illustration, we decompose the ACA-PC loss into two parts, *i.e.*, $\mathcal{L}_{\text{ACA-PC}} = \mathcal{L}_{\text{ali}} + \mathcal{L}_{\text{uni}}$. For the first part, $\mathcal{L}_{\text{ali}}$ serves as the alignment loss in traditional contrastive learning, which maximizes the inner product similarity between augmented samples from the same natural data:

$$\mathcal{L}_{\text{ali}} = 2\mathbb{E}_{\substack{\bar{x} \sim p(\bar{x}), x_i \sim p(x_i|\bar{x}) \\ x_j \sim p(x_j|\bar{x})}} - f_\theta(x_i)^\top f_\theta(x_j), \tag{11}$$

we use the mini-batch of natural sample to estimate $\mathbb{E}_{\bar{x} \sim p(\bar{x})}$. And we just use one sample to estimate $\mathbb{E}_{x_i \sim p(x_i|\bar{x})}$ and $\mathbb{E}_{x_j \sim p(x_j|\bar{x})}$ respectively. This leads to the traditional contrastive learning procedure : sample a mini-batch of natural data, augment it twice, compute and maximize the similarity of two augmented data.

For the second part, $\mathcal{L}_{\text{uni}}$ minimize the inner product similarity of augmented data from the marginal distribution:

$$\mathcal{L}_{\text{uni}} = N\mathbb{E}_{x_1 \sim p_\mathcal{A}(x_1), x_2 \sim p_\mathcal{A}(x_2)} \left[ \left( f_\theta(x_1)^\top f_\theta(x_2) \right)^2 \right]. \tag{12}$$

We use the in-batch augmented data to estimate $\mathbb{E}_{x_1 \sim p_\mathcal{A}(x_1)}$. Notably, two augmented samples randomly sampled are hardly augmented by the same natural sample. Therefore, following common practice (Chen et al., 2020a), we use two augmented data that are created by augmenting two different natural data to compute this term. Additionally, we find that $N$ in $\mathcal{L}_{\text{uni}}$ is too large to perform stable numerical computation. Thus in our implementation, we replace the $N$ with a tunable noise strength $K$.

For $\mathcal{L}_{\text{proj}}$, it is not efficient to fully sample from $p(x \mid \bar{x})$. However, it is notable that:

$$\mathcal{L}_{\text{proj}} = \mathbb{E}_{\bar{x} \sim p(\bar{x})} \left[ \| f_\theta(\bar{x}) - \mathbb{E}_{p(x|\bar{x})} f_\theta(x) \|_2^2 \right] = \mathbb{E}_{\substack{\bar{x} \sim p(\bar{x}), x_i \sim p(x_i|\bar{x}) \\ x_j \sim p(x_j|\bar{x})}} \left[ \| f_\theta(\bar{x}) - \frac{f_\theta(x_i) + f_\theta(x_j)}{2} \|_2^2 \right].$$

It has the same expectation subscript as $\mathcal{L}_{\text{ali}}$. So we can use the same strategy as $\mathcal{L}_{\text{ACA-PC}}$ and reuse the samples. $\mathcal{L}_{\text{proj}}$ is computed along with $\mathcal{L}_{\text{ali}}$ during training, *i.e.*, the principal component learning and projection are done simultaneously. That is why we call $\mathcal{L}_{\text{proj}}$ "on-the-fly projection".

The overall implementation of ACA is illustrated in Figure 4. And the algorithm is illustrated in Algorithm 1.

**Discussion on the relation with traditional contrastive learning.**    As is described in this section. ACA-PC takes a similar form as traditional contrastive learning methods. Similar to them, ACA-PC maximizes the inner product similarity of two views from the sample by Equation (11), and minimizes the squared inner product similarity of views from different samples. Note that we have proved that the learned embeddings by ACA-PC function as the principal components of the augmentation feature and preserve the posterior augmentation distances (Theorem 4.2). We believe traditional contrastive loss also has the similar functionality as ours. Due to the strong correlation

between augmentation overlap and semantic similarity, this may explain contrastive learning can learn semantically meaningful embeddings even though they ignore the semantic relationship between samples.

**Discussion on approximation in implementations.** There are several approximations to stabilize the training. First, we replace the factor $N$ in Equation (12) with a tunable noise strength $K$. Usually, the number of samples is very large in common datasets. When we use a complex model like DNN, it is unstable to involve such a large number in the loss. Therefore, we tune it small and find it works well. But we also note that we use $N$ in our synthetic experiment in Section 5 for the dimensionality and model is not too complex. The superior results proves effectiveness of our theory. Second, we normalize embeddings to project them into a sphere, which equals replacing the inner product with cosine similarity. We find this modification improves the performance from 81.84% to 91.58%.

---

**Algorithm 1** Augmentation Component Analysis Algorithm

---

**Require:** Unlabeled natural dataset $\{\boldsymbol{x}_i\}_{i=1}^{N}$; Augmentation function $\mathcal{A}$; Encoding model $f_\theta$ parameterized with $\theta$; projection parameter $\alpha$; Noise Strength $K$; Batch size $B$.
1: **for** sampled minibatch $\{\boldsymbol{x}_i\}_{i=1}^{B}$ **do**
2:     **for** $i = 1, 2, \ldots, B$ **do**
3:         $\boldsymbol{x}_i^{(1)} = \mathcal{A}(\bar{\boldsymbol{x}}_i), \boldsymbol{x}_i^{(2)} = \mathcal{A}(\bar{\boldsymbol{x}}_i)$
4:     **end for**
5:     $\mathcal{L}_{ACA-Full} = -\frac{2}{B}\sum_{i=1}^{B} f_\theta^\top(\boldsymbol{x}_i^{(1)})f_\theta(\boldsymbol{x}_i^{(2)}) + \frac{K}{B(B-1)}\sum_{i \neq j}(f_\theta^\top(\boldsymbol{x}_i^{(1)})f_\theta(\boldsymbol{x}_j^{(2)}))^2 +$
    $\alpha\|f_\theta(\bar{\boldsymbol{x}}) - \frac{f_\theta(\boldsymbol{x}_i^{(1)})+f_\theta(\boldsymbol{x}_i^{(2)})}{2}\|^2$
6:     update $\theta$ with w.r.t $\mathcal{L}_{ACA-Full}$.
7: **end for**
8: **return** $f_\theta$

---

## B EFFECT OF AUGMENTATION OVERLAPS

Like contrastive learning, our method relies on the quality of augmentation. Therefore, we investigate the influence of different augmentations and reveal the relationship between distribution difference and the linear probe performance on CIFAR10. The augmentation distribution is estimated by augmenting $10^6$ times for a subset of random 2000 pairs of samples with the number of intra-class and inter-class pairs being 1000 respectively. Note that as is stated in Section 4.1, even on CIFAR10, the actual value of $L$ is exponentially large (up to $256^{3072}$). It is impossible to accurately estimate a distribution over so many possible values. But we notice that for neural networks, many operators can reduce the possible number of values, like convolutions and poolings. Following this observation and to make the computation efficient, we descrete the color into 8-bit for each channel and use a max pooling operation to get a $4 \times 4$ picture. by this kind of approximation, the number of $L$ reduces to $8^{48}$. Seems still too large, but it can be noted that the augmentation distribution of each sample covers only a small region. It is enough to estimate the distribution by sampling. For memory restriction, we cannot fully estimate the weighted augmentation distance in Theorem 4.3. Because we cannot store all possible values for $p_{\mathcal{A}}(\boldsymbol{x})$. Instead, we use the Hellinger distance as the distribution distance measure:

$$\mathrm{d}_\mathrm{H}^2(\bar{\boldsymbol{x}}_1, \bar{\boldsymbol{x}}_2) = \frac{1}{N}\sum_{\boldsymbol{x} \in \mathcal{X}} \left(\sqrt{p(\boldsymbol{x} \mid \bar{\boldsymbol{x}}_1)} - \sqrt{p(\boldsymbol{x} \mid \bar{\boldsymbol{x}}_2)}\right)^2$$

Hellinger distance ranges $[0, 2]$, making the comparison clear.

We list the experimented augmentation here:

1. **Grayscale**: Randomly change the color into gray with probability of $0.1$.

2. **HorizontalFlip**: Randomly flip horizontally with probability $0.5$.

3. **Rotation**: Randomly rotate image with uniformly distributed angle in $[0, \pi]$.

4. **ColorJitter**: Jitter (brightness, contrast, saturation, hue) with strength $(0.4, 0.4, 0.4, 0.1)$ and probability $0.8$.

Table 3: Histogram (HIST) of distribution distances and linear probe accuracy (ACC) when using different augmentations on CIFAR10. Note that HIST is estimated in the input space. It is property of augmentation, regardless of learning algorithm. We aims to investigate the different augmentation overlaps caused by different augmentation, and reveal its connection between learned model. "Same" denotes the distance between samples with the same semantic class, and "Different" means different classes. The existence of overlap and relationship between intra-/inter-class distances affects the performance.

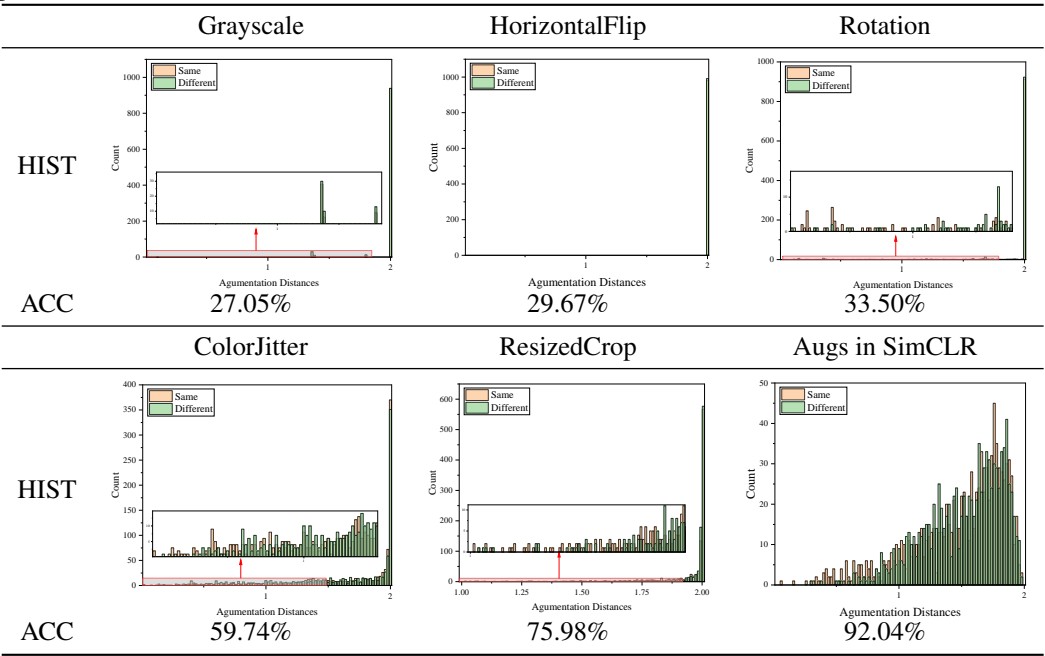

5. **ResizedCrop**: Extract crops with a random size from 0.2 to 1.0 of the original area and a random aspect ratio from 3/4 to 4/3 of the original aspect ratio.

6. **Augs in SimCLR**: Sequential combination of 5,4,1,2.

In Table 3, we display the histogram (HIST) of intra- and inter-class augmentation *distribution distances*. ACC displays the linear probe performance on the test set. From the table, the following requirements for a good augmentation can be concluded: (1) **Existence of overlap**. For the upper three augmentations. The "scope" of augmentation is small. As a result, most of the samples do not overlap. This makes embeddings lack the discriminative ability for downstream tasks. On the contrary, the lower three create overlaps for most of the samples, leading to much better performance. (2) **Intra-class distance is lower than inter-class**. Compared to ColorJitter, ResizedCrop makes more intra-class samples have lower distance. So ResizedCrop outperforms ColorJitter. SimCLR augmentation surpasses these two for the same reason. Interestingly, we find that the same phenomena appear when using other contrastive methods like SimCLR. It shows that these methods somehow utilize the augmentation overlap like our method.

## C  PERFORMANCE CURVE

In this section, we illustrate the performance curve throughout training. We aim to demonstrate the functionality of projection loss and show that our ACA method leads to better performance. The compared traditional contrastive learning method is chosen to be SimCLR, for the reason that *our method only differs from SimCLR in the loss, with all other things (architecture, optimizer and other shared hyperparameters) identical. Also, we do not introduce extra mechanisms like momentum encoder (BYOL, MoCo) and predictor (BYOL, SimSiam).*

Figure 5 shows the performance curve along with the projection loss on the CIFAR-10 dataset. The left figure shows the projection loss. We can see that in the early stage of training, the projection loss

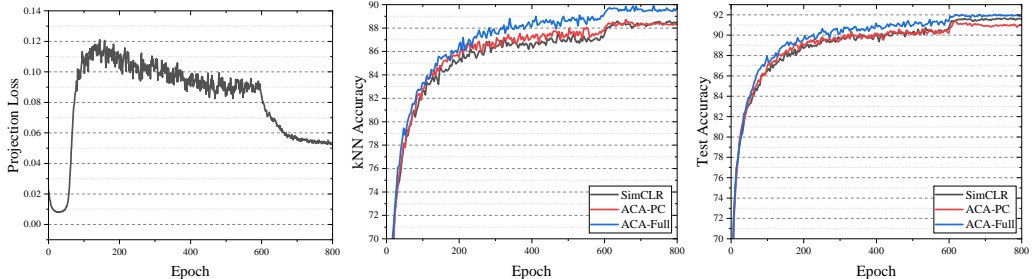

Figure 5: Values of projection loss and accuracy during training. Left is the projection loss. It rises in the early stage and is stable later, showing that the projection loss prevents the embeddings of natural data from deviating from the center of augmentation distribution. The mid and right figures show the performance curve measured by kNN accuracy and test accuracy. ACA-PC achieves similar performance as SimCLR but ACA-Full is better during the training process.

Table 4: We compare various SSL methods on transfer tasks by training linear layers. Only a single linear layer is trained on top of features. Simple random horizontal flips are used. Results except ours are taken from Koohpayegani et al. (2021). Our method can achieve competitive results with other contrastive learning methods despite short epochs, especially that it can surpass 1000-epoch SimCLR.

| Method | Epochs | Food | CIFAR10 | SUN | Cars | Aircraft | DTD | Pets | Caltech | Flowers | Mean |
|---|---|---|---|---|---|---|---|---|---|---|---|
| Supervised | | 72.30 | 93.60 | 61.90 | 66.70 | 61.00 | 74.90 | 91.50 | 94.50 | 94.70 | 78.90 |
| SimCLR | 1000 | 72.80 | 90.50 | 60.60 | 49.30 | 49.80 | **75.70** | 84.60 | 89.30 | 92.60 | 74.00 |
| MoCo v2 | 800 | 72.50 | **92.20** | 59.60 | 50.50 | 53.20 | 74.40 | 84.60 | 90.00 | 90.50 | 74.20 |
| BYOL | 1000 | **75.30** | 91.30 | **62.20** | **67.80** | **60.60** | 75.50 | **90.40** | **94.20** | **96.10** | **79.20** |
| BYOL-asym | 200 | 70.20 | 91.50 | 59.00 | 54.00 | 52.10 | 73.40 | 86.20 | 90.40 | 92.10 | 74.30 |
| MoCo v2 | 200 | 70.40 | 91.00 | 57.50 | 47.70 | 51.20 | **73.90** | 81.30 | 88.70 | 91.10 | 72.60 |
| MSF | 200 | 70.70 | 92.00 | 59.00 | **60.90** | 53.50 | 72.10 | **89.20** | 92.10 | **92.40** | **75.80** |
| MSF-w/s | 200 | 71.20 | **92.60** | **59.20** | 55.60 | 53.70 | 73.20 | 88.70 | **92.70** | 92.00 | 75.50 |
| ACA (ours) | 100 | **72.54** | 90.01 | **59.17** | 49.61 | **56.51** | 73.78 | 83.27 | 89.92 | 90.18 | 74.07 |

will increase. It reveals that the natural data will deviate from the center of augmentation distribution. It is harmful to the performance of the model. With the help of projection loss, the embeddings of natural data will be dragged back to their right position, the center. The mid and right figures illustrate the performance curve during training. With only ACA-PC loss, the model can only achieve similar performance during training. But the ACA-Full loss will help improve performance during training. Also, we can see that ACA starts to outperform SimCLR and ACA-PC by a considerable margin from about 50 epochs. This happens to be the epoch in which the projection loss increases to its stable level. Therefore, pulling the natural data to the center of its augmentation helps to learn better embeddings.

# D    TRANSFER TO OTHER DATASETS

Following Chen et al. (2020a), we evaluate the self-supervised pre-trained models for linear classification task on 10 datasets as it is conducted in MSF paper (Koohpayegani et al., 2021). The results are reported in Table 4. All the results other than ACA are taken from Koohpayegani et al. (2021). Although our method is trained with fewer epochs, it achieves competitive results with contrastive learning methods. Notably, it surpasses the 1000-epoch SimCLR which differs from our method only in loss. It shows that the embeddings learned by our method are also transferable to other downstream tasks. We think it is due to the universality of the correlation between augmentation similarity and semantical similarity across these benchmarks.

Table 5: We compare our models on the transfer task of object detction. We find that given a similar computational budget, our method is better than SimCLR, with shorter training time. The models are trained on the VOC `trainval07+12` set and evaluated on the `test07` set. We report average over 5 runs.

| Method | Epochs | $AP_{50}$ | AP | $AP_{75}$ |
|---|---|---|---|---|
| Sup. IN | - | 81.30 | 53.50 | 58.80 |
| Scratch | - | 60.20 | 33.80 | 33.10 |
| SimCLR (Chen et al., 2020a) | 200 | 81.80 | 55.50 | 61.40 |
| MoCo v2 (Chen et al., 2020c) | 200 | 82.30 | **57.00** | 63.30 |
| BYOL (Grill et al., 2020) | 200 | 81.40 | 55.30 | 61.10 |
| SwAV (Caron et al., 2020) | 200 | 81.50 | 55.40 | 61.40 |
| SimSiam (Chen & He, 2021) | 200 | **82.40** | **57.00** | **63.70** |
| MSF (Koohpayegani et al., 2021) | 200 | 82.20 | 56.70 | 63.40 |
| MSF w/s (Koohpayegani et al., 2021) | 200 | 82.20 | 56.60 | 63.10 |
| ACA-Full (ours) | 100 | 82.25 | 56.14 | 63.05 |

# E    TRANSFER TO OBJECT DETECTION

Following the procedure outlined in **?**, we use Faster-RCNN Ren et al. (2015) for the task of object detection on PASCAL-VOC Everingham et al. (2015). We use the code provided at MoCo repository[4] with default parameters. All the weights are finetuned on the `trainval07+12` set and evaluated on the `test07` set. We report an average over 5 runs in Table 5. Despite the shorter training epochs, our method can achieve better results than SimCLR, especially outperform by a large margin on $AP_{75}(> 1\%)$.

# F    PROOF OF LEMMA 4.1

For convenient, we define $M := \hat{A}^\top \hat{A}$. The elements of $M$ are:

$$M_{\boldsymbol{x}_1 \boldsymbol{x}_2} = \sum_{\bar{\boldsymbol{x}} \in \bar{\mathcal{X}}} \frac{p(\boldsymbol{x}_1 \mid \bar{\boldsymbol{x}}) p(\boldsymbol{x}_2 \mid \bar{\boldsymbol{x}})}{\sqrt{d_{\boldsymbol{x}_1}} \sqrt{d_{\boldsymbol{x}_2}}}, \boldsymbol{x}_1, \boldsymbol{x}_2 \in \mathcal{X} \tag{13}$$

Expanding Equation (3), we get:

$$
\begin{aligned}
\mathcal{L}_{mf} &= \sum_{\boldsymbol{x}_1, \boldsymbol{x}_2 \in \mathcal{X}} (M_{\boldsymbol{x}_1 \boldsymbol{x}_2} - F_{\boldsymbol{x}_1}^\top F_{\boldsymbol{x}_2})^2 \\
&= \sum_{\boldsymbol{x}_1, \boldsymbol{x}_2 \in \mathcal{X}} (M_{\boldsymbol{x}_1 \boldsymbol{x}_2} - \sqrt{d_{\boldsymbol{x}_1}} \sqrt{d_{\boldsymbol{x}_2}} f_\theta(\boldsymbol{x}_1)^\top f_\theta(\boldsymbol{x}_2))^2 \\
&= \text{const} - 2 \sum_{\boldsymbol{x}_1, \boldsymbol{x}_2 \in \mathcal{X}} \sqrt{d_{\boldsymbol{x}_1}} \sqrt{d_{\boldsymbol{x}_2}} M_{\boldsymbol{x}_1 \boldsymbol{x}_2} f_\theta(\boldsymbol{x}_1)^\top f_\theta(\boldsymbol{x}_2) + \sum_{\boldsymbol{x}_1, \boldsymbol{x}_2 \in \mathcal{X}} d_{\boldsymbol{x}_1} d_{\boldsymbol{x}_2} (f_\theta(\boldsymbol{x}_1)^\top f_\theta(\boldsymbol{x}_2))^2 \\
&= \text{const} - 2 \sum_{\boldsymbol{x}_1, \boldsymbol{x}_2 \in \mathcal{X}} \sum_{\bar{\boldsymbol{x}} \in \bar{\mathcal{X}}} p(\boldsymbol{x}_1 \mid \bar{\boldsymbol{x}}) p(\boldsymbol{x}_2 \mid \bar{\boldsymbol{x}}) f_\theta(\boldsymbol{x}_1)^\top f_\theta(\boldsymbol{x}_2) + \sum_{\boldsymbol{x}_1, \boldsymbol{x}_2 \in \mathcal{X}} d_{\boldsymbol{x}_1} d_{\boldsymbol{x}_2} (f_\theta(\boldsymbol{x}_1)^\top f_\theta(\boldsymbol{x}_2))^2
\end{aligned}
$$

---

[4]https://github.com/facebookresearch/moco

multiply by $p(\bar{x}) = \frac{1}{N}$ and replace $d_x$ with $\sum_{\bar{x}} p(x \mid \bar{x}) = Np_{\mathcal{A}}(x)$. The objective becomes:

$$
\begin{aligned}
\min_{\theta} \quad & -2 \sum_{x_1, x_2 \in \mathcal{X}} \sum_{\bar{x} \in \bar{\mathcal{X}}} p(x_1 \mid \bar{x}) p(x_2 \mid \bar{x}) p(\bar{x}) f_\theta(x_1)^\top f_\theta(x_2) \\
& + N \sum_{x_1, x_2 \in \mathcal{X}} p_{\mathcal{A}}(x_1) p_{\mathcal{A}}(x_2) (f_\theta(x_1)^\top f_\theta(x_2))^2 \\
= \; & -2 \mathbb{E}_{\bar{x} \sim p(\bar{x}), \; x_i \sim \mathcal{A}(x_i|\bar{x}) \atop x_j \sim \mathcal{A}(x_j|\bar{x})} \left[ f_\theta(x_1)^\top f_\theta(x_2) \right] \\
& + N \mathbb{E}_{x_1 \sim p_{\mathcal{A}}(x_1), x_2 \sim p_{\mathcal{A}}(x_2)} \left[ (f_\theta(x_1)^\top f_\theta(x_2))^2 \right] \\
= \; & \mathcal{L}_{\text{ACA-PC}}
\end{aligned}
$$

## G  PROOF OF THEOREM 4.2

As in Appendix F, we define $M := \hat{A}^\top \hat{A}$. By Eckart–Young–Mirsky theorem (Eckart & Young, 1936), the minimizer $\hat{F}$ of $\|M - FF^\top\|_F^2$, must have the form $\hat{V}\hat{\Sigma}Q$, where $\hat{V}, \hat{\Sigma}$ contain the top-k singular values and corresponding right singular vectors of $\hat{A}$, $Q \in \mathbb{R}^{k \times k}$ is some orthonormal matrix with $Q^\top Q = I$. Since we let $F_x = \sqrt{d_x} f_\theta(x)$, then the minimizer $\theta^\star$ must satisfy

$$
f_{\theta^\star}(x) = Q \frac{\hat{\boldsymbol{\sigma}} \odot \hat{\boldsymbol{v}}(x)}{\sqrt{d_x}} = Q \frac{[\sigma_1 v_1(x), \sigma_2 v_2(x), \dots, \sigma_k v_k(x)]^\top}{\sqrt{d_x}}.
$$

where $\odot$ is the element-wise multiplication. For convenience, we use $\sigma_i$ to denote $i$-th largest singular value, $u_i(\bar{x}), v_i(x)$ to denote the element of $i$-th left/right singular value corresponding to $\bar{x}/x$ . When $p(\bar{x}) = \frac{1}{N}$, $d_x = Np_{\mathcal{A}}(x) = \frac{p_{\mathcal{A}}(x)}{p(\bar{x})}$. Then the posterior distance:

$$
\begin{aligned}
\mathrm{d}_{\text{post}}^2(x_1, x_2) &= \sum_{\bar{x} \in \bar{\mathcal{X}}} \left( p_{\mathcal{A}}(\bar{x} \mid x_1) - p_{\mathcal{A}}(\bar{x} \mid x_2) \right)^2 \\
&= \sum_{\bar{x} \in \bar{\mathcal{X}}} \left( \frac{p(x_1 \mid \bar{x}) p(\bar{x})}{p_{\mathcal{A}}(x_1)} - \frac{p(x_1 \mid \bar{x}) p(\bar{x})}{p_{\mathcal{A}}(x_1)} \right)^2 \\
&= \sum_{\bar{x} \in \bar{\mathcal{X}}} \left( \frac{p(x_1 \mid \bar{x})}{d_{x_1}} - \frac{p(x_2 \mid \bar{x})}{d_{x_2}} \right)^2 \\
&= \sum_{\bar{x} \in \bar{\mathcal{X}}} \left( \frac{\hat{A}_{\bar{x}x_1}}{\sqrt{d_{x_1}}} - \frac{\hat{A}_{\bar{x}x_2}}{\sqrt{d_{x_2}}} \right)^2 \\
&= \sum_{\bar{x} \in \bar{\mathcal{X}}} \left( \sum_{i=1}^{N} \frac{\sigma_i u_i(\bar{x}) v_i(x_1)}{\sqrt{d_{x_1}}} - \frac{\sigma_i u_i(\bar{x}) v_i(x_2)}{\sqrt{d_{x_2}}} \right)^2 \\
&= \sum_{\bar{x} \in \bar{\mathcal{X}}} \left( \sum_{i=1}^{N} \sigma_i u_i(\bar{x}) \left( \frac{v_i(x_1)}{\sqrt{d_{x_1}}} - \frac{v_i(x_2)}{\sqrt{d_{x_2}}} \right) \right)^2 \\
&= \sum_{\bar{x} \in \bar{\mathcal{X}}} \sum_{i, i'} \sigma_i u_i(\bar{x}) \sigma_{i'} u_{i'}(\bar{x}) \left( \frac{v_i(x_1)}{\sqrt{d_{x_1}}} - \frac{v_i(x_2)}{\sqrt{d_{x_2}}} \right) \left( \frac{v_{i'}(x_1)}{\sqrt{d_{x_1}}} - \frac{v_{i'}(x_2)}{\sqrt{d_{x_2}}} \right) \\
&= \sum_{i, i'} \sigma_i \sigma_{i'} \left( \frac{v_i(x_1)}{\sqrt{d_{x_1}}} - \frac{v_i(x_2)}{\sqrt{d_{x_2}}} \right) \left( \frac{v_{i'}(x_1)}{\sqrt{d_{x_1}}} - \frac{v_{i'}(x_2)}{\sqrt{d_{x_2}}} \right) \sum_{\bar{x} \in \bar{\mathcal{X}}} u_i(\bar{x}) u_{i'}(\bar{x}) \\
&\overset{(1)}{=} \sum_{i, i'} \sigma_i \sigma_{i'} \left( \frac{v_i(x_1)}{\sqrt{d_{x_1}}} - \frac{v_i(x_2)}{\sqrt{d_{x_2}}} \right) \left( \frac{v_{i'}(x_1)}{\sqrt{d_{x_1}}} - \frac{v_{i'}(x_2)}{\sqrt{d_{x_2}}} \right) \delta_{i, i'} \\
&= \sum_{i=1}^{N} \sigma_i^2 \left( \frac{v_i(x_1)}{\sqrt{d_{x_1}}} - \frac{v_i(x_2)}{\sqrt{d_{x_2}}} \right)^2
\end{aligned}
$$

(14)

(1) is due to the orthogonality of singular vectors. Note that:

$$\sum_{i=1}^{N}(\frac{v_i(\boldsymbol{x_1})}{\sqrt{d_{\boldsymbol{x_1}}}} - \frac{v_i(\boldsymbol{x_2})}{\sqrt{d_{\boldsymbol{x_2}}}})^2$$

$$=\sum_{i=1}^{L}(\frac{v_i(\boldsymbol{x_1})}{\sqrt{d_{\boldsymbol{x_1}}}} - \frac{v_i(\boldsymbol{x_2})}{\sqrt{d_{\boldsymbol{x_2}}}})^2 - \sum_{i=N+1}^{L}(\frac{v_i(\boldsymbol{x_1})}{\sqrt{d_{\boldsymbol{x_1}}}} - \frac{v_i(\boldsymbol{x_2})}{\sqrt{d_{\boldsymbol{x_2}}}})^2$$

$$\leq\sum_{i=1}^{L}(\frac{v_i(\boldsymbol{x_1})}{\sqrt{d_{\boldsymbol{x_1}}}} - \frac{v_i(\boldsymbol{x_2})}{\sqrt{d_{\boldsymbol{x_2}}}})^2$$

$$=\sum_{i=1}^{L}\frac{v_i^2(\boldsymbol{x_1})}{d_{\boldsymbol{x_1}}} + \sum_{i=1}^{L}\frac{v_i^2(\boldsymbol{x_2})}{d_{\boldsymbol{x_2}}} - 2\sum_{i=1}^{L}\frac{v_i(\boldsymbol{x_1})v_i(\boldsymbol{x_2})}{\sqrt{d_{\boldsymbol{x_1}}}\sqrt{d_{\boldsymbol{x_2}}}}$$

$$=\frac{1}{d_{\boldsymbol{x_1}}} + \frac{1}{d_{\boldsymbol{x_2}}} - \frac{2\delta_{\boldsymbol{x_1}\boldsymbol{x_2}}}{\sqrt{d_{\boldsymbol{x_1}}}\sqrt{d_{\boldsymbol{x_2}}}}$$

$$\overset{(2)}{\leq}(\frac{1}{d_{\boldsymbol{x_1}}} + \frac{1}{d_{\boldsymbol{x_2}}})(1 - \delta_{\boldsymbol{x_1}\boldsymbol{x_2}})$$

$$\leq\frac{2}{d_{\min}}(1 - \delta_{\boldsymbol{x_1}\boldsymbol{x_2}})$$

(2) can be deduced by considering conditions whether $\boldsymbol{x_1} = \boldsymbol{x_2}$ or not. Then:

$$\|f_{\theta^\star}(\boldsymbol{x_1}) - f_{\theta^\star}(\boldsymbol{x_2})\|^2$$

$$=\sum_{i=1}^{k}\sigma_i^2(\frac{v_i(\boldsymbol{x_1})}{\sqrt{d_{\boldsymbol{x_1}}}} - \frac{v_i(\boldsymbol{x_2})}{\sqrt{d_{\boldsymbol{x_2}}}})^2$$

$$=\mathrm{d}_{\text{post}}^2(\boldsymbol{x_1}, \boldsymbol{x_2}) - \sum_{i=k}^{N}\sigma_i^2(\frac{v_i(\boldsymbol{x_1})}{\sqrt{d_{\boldsymbol{x_1}}}} - \frac{v_i(\boldsymbol{x_2})}{\sqrt{d_{\boldsymbol{x_2}}}})^2 \quad (\leq \mathrm{d}_{\text{post}}^2(\boldsymbol{x_1}, \boldsymbol{x_2}))$$

$$\geq\mathrm{d}_{\text{post}}^2(\boldsymbol{x_1}, \boldsymbol{x_2}) - \sigma_{k+1}^2\sum_{i=k+1}^{N}(\frac{v_i(\boldsymbol{x_1})}{\sqrt{d_{\boldsymbol{x_1}}}} - \frac{v_i(\boldsymbol{x_2})}{\sqrt{d_{\boldsymbol{x_2}}}})^2$$

$$\geq\mathrm{d}_{\text{post}}^2(\boldsymbol{x_1}, \boldsymbol{x_2}) - \sigma_{k+1}^2\sum_{i=1}^{N}(\frac{v_i(\boldsymbol{x_1})}{\sqrt{d_{\boldsymbol{x_1}}}} - \frac{v_i(\boldsymbol{x_2})}{\sqrt{d_{\boldsymbol{x_2}}}})^2$$

$$\geq\mathrm{d}_{\text{post}}^2(\boldsymbol{x_1}, \boldsymbol{x_2}) - \frac{2\sigma_{k+1}^2}{d_{\min}}(1 - \delta_{\boldsymbol{x_1}\boldsymbol{x_2}})$$

Therefore, we have proved Theorem 4.2.

## H    PROOF OF THEOREM 4.3

similar to Appendix G,

$$d_{\text{w-aug}}^2(\bar{\boldsymbol{x}}_1, \bar{\boldsymbol{x}}_2) = \sum_{\boldsymbol{x} \in \mathcal{X}} \frac{1}{N p_{\mathcal{A}}(\boldsymbol{x})} \left( p(\boldsymbol{x} \mid \bar{\boldsymbol{x}}_1) - p(\boldsymbol{x} \mid \bar{\boldsymbol{x}}_2) \right)^2$$

$$= \sum_{\boldsymbol{x} \in \mathcal{X}} \left( \frac{p(\boldsymbol{x} \mid \bar{\boldsymbol{x}}_1)}{\sqrt{N p_{\mathcal{A}}(\boldsymbol{x})}} - \frac{p(\boldsymbol{x} \mid \bar{\boldsymbol{x}}_1)}{\sqrt{N p_{\mathcal{A}}(\boldsymbol{x})}} \right)^2$$

$$= \sum_{\boldsymbol{x} \in \mathcal{X}} \left( \frac{p(\boldsymbol{x} \mid \bar{\boldsymbol{x}}_1)}{\sqrt{d_{\boldsymbol{x}}}} - \frac{p(\boldsymbol{x} \mid \bar{\boldsymbol{x}}_1)}{\sqrt{d_{\boldsymbol{x}}}} \right)^2$$

$$= \sum_{\boldsymbol{x} \in \mathcal{X}} \left( \hat{A}_{\bar{\boldsymbol{x}}_1 \boldsymbol{x}} - \hat{A}_{\bar{\boldsymbol{x}}_2 \boldsymbol{x}} \right)^2$$

$$= \sum_{\boldsymbol{x} \in \mathcal{X}} \left( \sum_{i=1}^N \sigma_i u_i(\bar{\boldsymbol{x}}_1) v_i(\boldsymbol{x}) - \sigma_i u_i(\bar{\boldsymbol{x}}_2) v_i(\boldsymbol{x}) \right)^2$$

$$= \sum_{\boldsymbol{x} \in \mathcal{X}} \left( \sum_{i=1}^N \sigma_i (u_i(\bar{\boldsymbol{x}}_1) - u_i(\bar{\boldsymbol{x}}_2)) v_i(\boldsymbol{x}) \right)^2$$

$$= \sum_{\boldsymbol{x} \in \mathcal{X}} \sum_{i,i'} \sigma_i v_i(\boldsymbol{x}) \sigma_{i'} v_{i'}(\boldsymbol{x}) (u_i(\bar{\boldsymbol{x}}_1) - u_i(\bar{\boldsymbol{x}}_2)) (u_{i'}(\bar{\boldsymbol{x}}_1) - u_{i'}(\bar{\boldsymbol{x}}_2))$$

$$= \sum_{i,i'} \sigma_i \sigma_{i'} (u_i(\bar{\boldsymbol{x}}_1) - u_i(\bar{\boldsymbol{x}}_2)) (u_{i'}(\bar{\boldsymbol{x}}_1) - u_{i'}(\bar{\boldsymbol{x}}_2)) \sum_{\boldsymbol{x} \in \mathcal{X}} v_i(\boldsymbol{x}) v_{i'}(\boldsymbol{x})$$

$$\overset{(1)}{=} \sum_{i,i'} \sigma_i \sigma_{i'} (u_i(\bar{\boldsymbol{x}}_1) - u_i(\bar{\boldsymbol{x}}_2)) (u_{i'}(\bar{\boldsymbol{x}}_1) - u_{i'}(\bar{\boldsymbol{x}}_2)) \delta_{i,i'}$$

$$= \sum_{i=1}^N \sigma_i^2 (u_i(\boldsymbol{x}_1) - u_i(\boldsymbol{x}_2))^2$$

(1) is due to the orthogonality of singular vectors. And $g(\bar{\boldsymbol{x}})$ takes the following form:

$$g(\bar{\boldsymbol{x}}) = Q \left[ \sigma_1^2 u_1(\boldsymbol{x}), \sigma_2^2 u_2(\boldsymbol{x}), \dots, \sigma_k^2 u_k(\boldsymbol{x}) \right]^\top.$$

Thus,

$$\|g(\bar{\boldsymbol{x}}_1) - g(\bar{\boldsymbol{x}}_2)\|_{\Sigma_k^{-2}}^2 = \sum_{i=1}^k \sigma_i^2 (u_i(\boldsymbol{x}_1) - u_i(\boldsymbol{x}_2))^2$$

$$= d_{\text{w-aug}}^2(\bar{\boldsymbol{x}}_1, \bar{\boldsymbol{x}}_2) - \sum_{i=k+1}^N \sigma_i^2 (u_i(\boldsymbol{x}_1) - u_i(\boldsymbol{x}_2))^2 \quad (\leq d_{\text{w-aug}}^2(\bar{\boldsymbol{x}}_1, \bar{\boldsymbol{x}}_2))$$

$$\geq d_{\text{w-aug}}^2(\bar{\boldsymbol{x}}_1, \bar{\boldsymbol{x}}_2) - \sigma_{k+1}^2 \sum_{i=1}^N (u_i(\boldsymbol{x}_1) - u_i(\boldsymbol{x}_2))^2$$

$$= d_{\text{w-aug}}^2(\bar{\boldsymbol{x}}_1, \bar{\boldsymbol{x}}_2) - 2\sigma_{k+1}^2 (1 - \delta_{\bar{\boldsymbol{x}}_1 \bar{\boldsymbol{x}}_2})$$

## I  ABLATION STUDY ON PARAMETER $\alpha$ AND $K$

We conduct ablation experiments on the parameter $\alpha$ and $K$. $\alpha$ is the trade-off parameter between ACA-PC loss and projection loss Equation (10). $K$ act as the noise strength for ACA-PC, which replaces $N$ in Equation (4).

Figure 6 shows the effect of $\alpha$ and $K$ on different benchmarks. It can be seen that $\alpha$ is necessary to improve the performance of ACA-PC. A certain value of $\alpha$ helps the model to achieve better results. However, a too large value of $\alpha$ degrades the performance. The same phenomenon is the same on $K$.

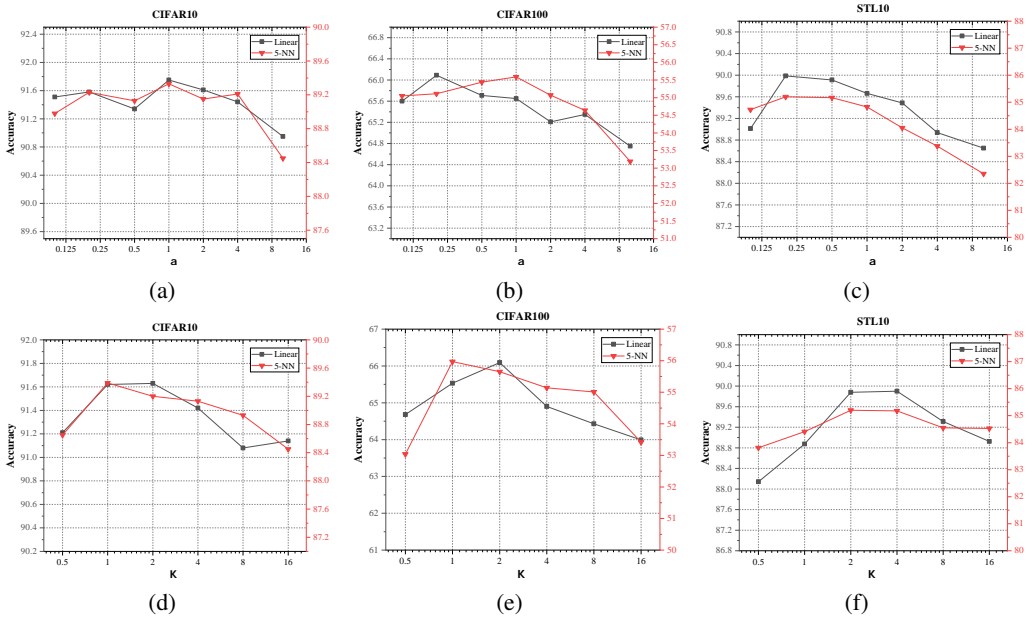

Figure 6: Ablation studies on the effect of $\alpha$ and $K$. We report linear classification and 5-nearest neighbor accuracy on different datasets with the ResNet-18 encoder. The upper 3 figures illustrate the effect of $\alpha$ on 3 different datasets. The lower 3 figures illustrate the performance v.s. $K$.

## J    COMPARISON OF NEAREST NEIGHBORS

We randomly select 8 samples from the validation set of ImageNet-100 (Tian et al., 2020a). Then we use the encoder learned by our ACA method and SimCLR (Chen et al., 2020a) to extract features and investigate their nearest neighbors of them. The left-most column displays the selected samples and the following columns show the 5 nearest neighbors. The samples labeled as different classes are marked by the red box. We also annotate the distance between the samples and their nearest neighbors. First, we can see that even though utilizing the augmentation in a different way, ACA achieves similar results as traditional contrastive learning. Both of them can learn semantically meaningful embeddings. However, we can see that ACA tends to learn embeddings that pull together images that are similar in the input space, *i.e.*, creating similar augmentation, while SimCLR sometimes has neighbors that seem different.

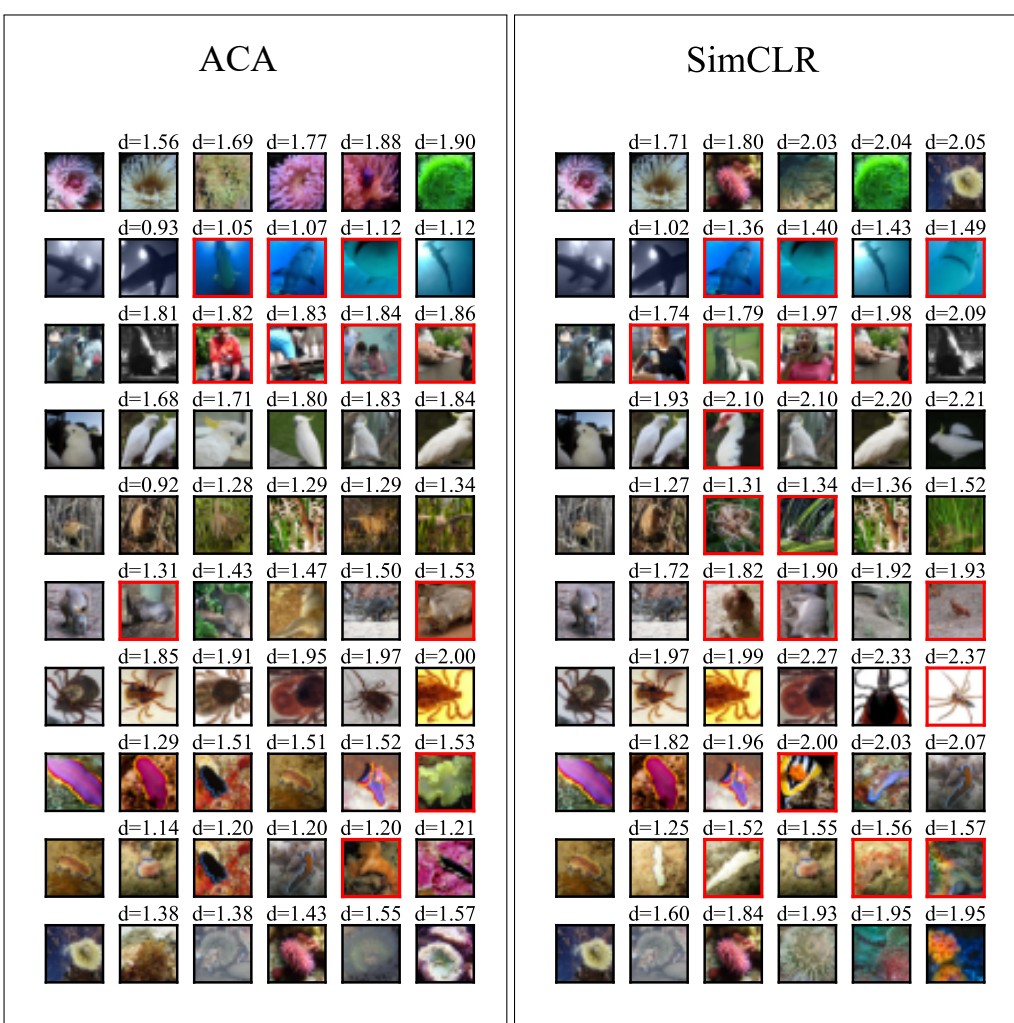

Figure 7: The 5 nearest neighbors of selected samples in the embedding space of ACA and SimCLR. The images were taken from the ImageNet-100 validation set. Distances between selected samples and their nearest neighbors are annotated above each picture. We can see that the embeddings learned by ACA tend to pull together images that are similar in the input space, *i.e.*, creating similar augmentation. While SimCLR sometimes has neighbors that seem different.

