# OpenReview forum: "Augmentation Component Analysis: Modeling Similarity via the Augmentation Overlaps"
_ICLR.cc/2023/Conference — ICLR 2023 poster_

### Official Review · Reviewer_gr8r · 2022-10-21

**Confidence:** 4
**Correctness:** 4
**Technical Novelty And Significance:** 3
**Empirical Novelty And Significance:** 2
**Recommendation:** 8

**Clarity, Quality, Novelty And Reproducibility:**

The clarity is okay, there does exist room for improvement.

The quality is above average.

The novelty is okay, there do exist novel theoretical contributions, but neither the motivation nor the presented method is super novel.


**Strength And Weaknesses:**

Strength
- The proposed method is theoretically sound and complete. The deduced loss function seems new and interesting.
- The paper provided interesting pilot study and also interesting small experiments for motivation purposes.
- The evaluation experiments are in general complete and nicely conducted. The method is validated on many datasets, with sufficient benchmarking.


Weaknesses
1. Major concern: model performance
- The model performance on several datasets seems to be lower than previously reported results. Specifically, check [1, 2] where many numbers are greater than the presented method. Unfortunately, as the major experimental result of the presented method is framed to be about surpassing previous state-of-the-art (that is the only experiment), the presented numbers are just not satisfactory enough, which greatly weakens the paper’s contribution.

2. The related works and the references lack more recent works.
- There are many contrastive learning/self-supervised learning methods in 2021 and 2022 that are not included in related works. For example, check [2-5].

3. The writing and organization of the paper could be improved.
- The abstract could be improved. Specifically, I suggest rewording the first sentence and the 4th sentence.
- Section 4.1 and section 4.2 are a bit wordy and might be simplified. The computation overload is mentioned many times in two adjacent sections.
- Some of the preliminary in section 3 could be moved into related works, especially the second paragraph, which has much overlapping information with section 2.


4. Minor comments:
- I suggest rewording Section 3 first sentence.
- Related works the Self-Supervised learning section should mention non-contrastive methods as well, like BYOL. Instead, the paper put non-contrastive methods inside the contrastive learning section and implicitly infer them as Contrastive Learning methods without using negative samples, which is a bit controversial because BYOL is not always referred to as contrastive methods though some studies suggest that they intrinsically are.
- What is the connection between this method and Barlow Twins?
- It might be more beneficial if the authors could explore other use cases of the presented learning method rather than just performing classification. For example, if the proposed loss function could provide an “augmentation spectrum” of certain data, could that provide more informative latent space based on some other metrics when compared to other methods?


[1] Ermolov, Aleksandr, et al. "Whitening for self-supervised representation learning." International Conference on Machine Learning. PMLR, 2021.

[2] Azabou, Mehdi, et al. "Mine your own view: Self-supervised learning through across-sample prediction." arXiv preprint arXiv:2102.10106 (2021).

[3] Zbontar, Jure, et al. "Barlow twins: Self-supervised learning via redundancy reduction." International Conference on Machine Learning. PMLR, 2021.

[4] Tsai, Yao-Hung Hubert, et al. "A note on connecting barlow twins with negative-sample-free contrastive learning." arXiv preprint arXiv:2104.13712 (2021).

[5] Kalantidis, Yannis, et al. "Hard negative mixing for contrastive learning." Advances in Neural Information Processing Systems 33 (2020): 21798-21809.


**Summary Of The Paper:**

The paper presented a self-supervised learning method based on the augmentation distribution of the image. Specifically, the paper is motivated by the intuition that semantically similar images tend to have similar augmented views, and proposed a method called ACA to learn the augmentation principal components of data. The method is theoretically deduced to maintain and measure the discrepancy between the augmentation distribution across samples. The method is validated on many benchmarks (cifar, stl, imagenet), and shows competitive performance when compared with several existing models.

**Summary Of The Review:**

I think this paper is borderline and I lean toward borderline rejection. The paper is based on valid motivation, the method seems to be relatively novel and sound, and the experiments are in general nicely organized. However, the paper lacks sufficient experimental results that support the superiority of the presented method, as the majority of the reported performance is lower than previously reported results. Unfortunately, this is the majority component of the experiments section, and as the paper is framed to present a method that could surpass/ provide comparable performance w.r.t existing methods, the experimental results outplay the methodology contribution imho.

---

> ### Author Response · Authors · 2022-11-15
> **Reply to Reviewer gr8r**
>
> We are thankful for your precious advice and acknowledgment of our contributions.
>
> **Q1**: Major concern: model performance
>
> **A1**:
> Thanks for pointing out the difference between the results from other papers. We find that it is attributed to the forward pass (perhaps influencing BN) and the evaluation protocol. But some of the results still cannot be reproduced, like the results on Tiny-ImageNet and STL10. **We have provided the code, and all the methods run under the same hyperparameters**. ACA is re-evaluated, and it still achieves competitive results. (By now, not all the results are reproduced, so we temporally do not change the reported number.) Additionally, we provide the performance curve in comparison to SimCLR. We choose SimCLR because our method only differs from SimCLR in the loss, with all other things (architecture, optimizer) identical. Also, we do not introduce extra mechanisms like momentum encoder (BYOL, MoCo) and predictor(BYOL, SimSiam). This experiment is illustrated in figure 3 in the revised paper and shows the superiority of our method.
>
> We do not take performance as the main contribution of our work. We are the first to view the augmentation distribution as a feature of a sample. And we propose an efficient method ACA to perform PCA on this feature. ACA theoretically preserves the similarity of augmentation distribution, which is reasonable based on the observation that semantically similar samples create similar views. We think the idea of this paper should outweigh the performance.
>
> We conduct the experiment with the following purposes (1) **the motivation is correct**. In the pilot study (Now moved to appendix B), we show **the sample pairs from the same class have smaller augmentation distances than pairs from different classes** (Figure 5 left). Also, the **augmentation feature itself is a discriminative representation** (Figure 5 right, the AF line). In Appendix C, we show that **how views of samples overlap influence the quality of learned embeddings**. (2) **the method is effective**. In the pilot study and experiments on real-world datasets, our ACA method achieves competitive and better results against traditional contrastive learning methods without introducing extra mechanisms like mining nearest neighbors, mixup, momentum encoder, or memory banks. It should be enough to show the effectiveness of the loss. (3) **The resemblance to contrastive learning in loss form and performance may help explain the success of contrastive learning**.
>
> ---
>
> **Q2**: The related works and the references lack more recent works. Related works the Self-Supervised learning section should mention non-contrastive methods as well
>
> **A2**: We have reorganized the related work of contrastive learning and cited the missed papers.
>
> ---
>
> **Q3**: The writing and organization of the paper could be improved. I suggest rewording Section 3 first sentence.
>
> **A3**: Thanks very much for the valuable suggestions. We have improved the organization to make it more clear.
>
> ---
>
> **Q4**: What is the connection between this method and Barlow Twins?
>
> **A4**: It is an interesting question. We thank the authors of [1] for providing an inspiring interpretation of Barlow Twins. In our opinion, the redundancy reduction in Barlow Twins has some connection to the learning of eigenvectors in PCA since the eigenvectors are required to be orthonormal. However, the idea of Barlow Twins is very different from ours, and we are not sure if there is a strong connection. We are happy to discuss it :).
>
> ---
>
> **Q5**: It might be more beneficial if the authors could explore other use ... some other metrics when compared to other methods?
>
> **A5**: We tried to compute some statistics about the augmentation feature but found it very hard, even on small-scaled datasets like CIFAR10. Instead, we present the histogram of augmentation distances and the quality of embedding learned by applying different augmentation methods in Appendix C. Of course, the augmentation distance is computed in an approximated way. We find that the overlap matters much for learning good representations. The results validate our motivation and method – our ACA preserves the augmentation similarity.
>
>
> ---
>
> We are looking forward to seeing the change in your decision if we have made things clear :).
>
>
> [1] Tsai, Yao-Hung Hubert, et al. "A note on connecting barlow twins with negative-sample-free contrastive learning." arXiv preprint arXiv:2104.13712 (2021).

---

> > ### Comment · Reviewer_gr8r · 2022-11-15
> > **Question followup**
> >
> > "We find that it is attributed to the forward pass (perhaps influencing BN) and the evaluation protocol.": would you further elaborate?

---

> > > ### Author Response · Authors · 2022-11-15
> > > **Reply**
> > >
> > > In the original implementation, we concatenate the views and forward them through the network. We then split them in the embedding space.   Now we find separately forwarding them leads to better results. Also, the 100 epochs training of the linear classifier may not be enough. We now choose a 500-epoch training. Details can be found in the code.

---

> ### Comment · Reviewer_gr8r · 2022-11-16
> **Increase score**
>
> After reviewing the authors' rebuttal and the revised paper, I decided to increase the score from 5 to 8. I am impressed by the revision (it addressed the majority of my concerns) and the additional experiments that the authors provided (mostly appendix D, I would encourage the authors to merge these results (the table) in the main text, ideally with more analysis). With that being said, I had a hard time deciding if I should give a score 8 or 6 (I would give a score of 7 if the option is available). My suggestion for this paper is at the midpoint between borderline and accept.
>
> To be more clear, my decision is mostly based on: (1) I am satisfied with the theoretical/methodology contribution previously, with (previous) major concern on experiments, as there is only one experiment and the results are not super impressive. (2) The authors provided more experiments on transfer, and provided more analysis in Section 5.3, which ameliorated my major concern by a large margin. I do think more improvements could be done to improve the experiments part, but I think the provided amount is sufficient (accompanied by other contributions). (3) The authors addressed my other concerns carefully, which further improved the paper's quality.

---

> > ### Author Response · Authors · 2022-11-17
> > **Thanks for increasing score**
> >
> > We are very thankful for increasing the score. It is also very helpful to us through the discussion.

---

### Official Review · Reviewer_1C2x · 2022-10-22

**Confidence:** 3
**Correctness:** 4
**Technical Novelty And Significance:** 3
**Empirical Novelty And Significance:** 3
**Recommendation:** 6

**Clarity, Quality, Novelty And Reproducibility:**

- Clarity: The manuscript has some clarity issues as commented above, but overall it is written clearly.
- Reproducibility: The paper elaborates on technical details for reproducing the reported results.
- Novelty: The main idea of this paper is new and validated both theoretically and empirically.

Overall, I believe the paper meets the standard of ICLR in terms of quality.


**Strength And Weaknesses:**

[Strengths]
- The main idea about relations between semantic similarity and similarity between augmentation distributions is new in self-supervised representation learning and sounds reasonable. The idea also has been justified empirically.
- The way of simulating the direct yet impractical implementation of the idea using a neural network and two loss functions is interesting, solid (its validity is proven in the appendix), and worth to be introduced to the community.
- The proposed method achieved best scores on multiple benchmarks for self-supervised representation learning.


[Weaknesses]
- Clarity issues: (1) The third paragraph of Section 1 is hard to grasp; it will be useful if a figure that conceptually illustrates the main idea is added. (2) The notion of augmented sample x is not clear and seems not consistent in Section 3 and 4. In Section 3, x is introduced as if it is a new view of a natural image (i.e., an augmented version of the natural image), but in Section 4 it is coupled with all natural images to form posterior probabilities.
- Lack of experiments: The proposed method needs to be compared with latest approaches to self-supervised representation learning. Also, I would recommend evaluating the proposed method on other epochs (e.g., 200, 400, 800) like prior work for the ImageNet linear classification benchmark. Moreover, advantages of the learned model have to be also validated for downstream tasks other than image classification, i.e., transfer learning setting for object detection and instance segmentation.


**Summary Of The Paper:**

This paper presents a new method for self-supervised representation learning based on data augmentation. Unlike prior work in this line of research, the proposed method considers semantic similarity (e.g., class equivalence) between different instances as well as semantic invariance between different views of the same instance. The key idea is that if two training instances are semantically close to each other, their augmented versions (i.e., different views) will be largely overlapped. Conceptually, the idea can be implemented by comparing posterior probabilities of augmented examples given original ones, called augmentation features, which however is computationally intractable due to the prohibitively large number of potential augmented examples and the difficulty of estimating such probabilities. The authors thus propose to simulate principal component analysis (PCA) of augmentation features through a neural network encoder trained by two loss functions; they also prove that the loss functions allow the encoder to produce PCA projection results without computing the augmentation features explicitly. The proposed method outperforms prior work in the same direction and provides theoretical interpretation of contrastive methods for self-supervised representation learning.

**Summary Of The Review:**

The paper presents a new idea for self-supervised representation learning, which enables us to consider both semantic similarity between different instances and semantic invariance between different views of the same instance. The implementation of the idea is solid theoretically and achieves outstanding performance on multiple public benchmarks. These positive points outweigh my concerns on lack of experiments, and thus I believe the paper is worthy to be introduced to the community.

---

> ### Author Response · Authors · 2022-11-15
> **Reply to Reviewer 1C2x**
>
> We are very thankful for your acknowledgment of our work! Following are our responses to the concerns.
>
> ---
>
> **Q1**: The third paragraph of Section 1 is hard to grasp; it will be useful if a figure that conceptually illustrates the main idea is added.
>
> **A1**: The main idea of this paragraph is illustrated mainly in figure 2. The augmentation feature is the ideal representation that can model augmentation similarity. But it is extremely high-dimensional and thus impractical to use. So we need to perform dimension reduction on it.
>
> ---
>
> **Q2**:  The notion of augmented sample x is not clear and seems not consistent in Section 3 and 4. In Section 3, x is introduced as if it is a new view of a natural image (i.e., an augmented version of the natural image), but in Section 4 it is coupled with all natural images to form posterior probabilities.
>
> **A2**: We use $\bar{x}$ to represent the natural data, and $x$ to represent views of natural data. The two distributions $p(x|\bar{x})$ and $p_{\mathcal{A}}(\bar{x}|x)$ have different meaning. $p(x|\bar{x})$ means the probability of getting $x$ by augmenting $\bar{x}$. $p_{\mathcal{A}}(\bar{x}|x)$ means the probability that $x$ is augmented by $\bar{x}$. $p_{\mathcal{A}}(\bar{x}|x)$ is the posterior distribution of $p(x|\bar{x})$. We use the subscript $\mathcal{A}$ to distinguish them to avoid confusion. Are there any mistakes that confuse you?
>
> ---
>
> **Q3**: Lack of experiments.
>
> **A3**: We apologize that we are not able to finish ImageNet training by now. Training on ImageNet is too costly for our device. We need one week to train a ResNet-50 model for 100 epochs. Currently, we can only provide the result of 100 epochs. But we will make up for the results of the following epochs. As compensation, we provide results of transfer learning on 10 benchmarks (Table 3 in Section 5.4) and the object detection results (Table 5 in Appendix D) with the 100-epoch checkpoint. Despite shorter training epochs, our method shows competitive results with other contrastive learning methods.

---

> > ### Comment · Reviewer_1C2x · 2022-11-27
> > **Post-rebuttal comment**
> >
> > Many thanks for the response and the revision. I still champion this submission, but do not further increase the score since my major concern (lack of experiments) has not been fully resolved. Also, I still believe that Section 1 of the paper should be carefully revised to deliver the key idea more clearly (I do not think this is a critical issue though): the notion of "augmentation feature", which is firstly introduced in the section, is hard to grasp unless one reads Section 3 and 4 beforehand. One way to resolve the issue would be referring to Figure 2 in the third paragraph of Section 1, but the figure is unfortunately not intuitive enough.

---

> > > ### Author Response · Authors · 2022-11-27
> > > **Reply to post-rebuttal comment**
> > >
> > > Thanks very much for the precious suggestions. We decide to change the description of this paragraph.
> > >
> > > Before:
> > > > The augmentation overlap naturally acts as the self-supervision to learn embeddings.  Following this idea, we can manually construct an ``ideal'' (but possibly infinite-dimensional) *augmentation feature* for a sample --- whose elements reveal the probability of augmenting that sample to another and encode the augmentation information comprehensively. However, not only are its elements hard to calculate, but also such high-dimensional embeddings are impractical to use.
> > >
> > > After:
> > > > In this paper, we propose to describe data directly by their augmentation distributions. We call a feature of this kind the *augmentation feature*. The elements of the augmentation feature represent the probability of getting a certain view by augmenting the sample as shown in the left of Figure 2. The augmentation feature serves as an ``ideal'' representation since it encodes the augmentation information without any loss and we can easily obtain the overlap of two samples from it. However, not only are its elements hard to calculate, but also such high-dimensional embeddings are impractical to use.
> > >
> > > ---
> > >
> > > We are also thankful for championing our work.

---

### Official Review · Reviewer_fbkG · 2022-10-26

**Confidence:** 4
**Correctness:** 4
**Technical Novelty And Significance:** 2
**Empirical Novelty And Significance:** 2
**Recommendation:** 6

**Clarity, Quality, Novelty And Reproducibility:**

The code and pre-trained models are desired. After reading this paper, it is not easy for me to reproduce the results. I acknowledge the novelty of this paper, but the poor experimental results need to be more convincing.

**Strength And Weaknesses:**

**Strength**
1. The idea of augmentation component analysis is novel and interesting.
2. The theoretical analysis of this work is promising.

**Weaknesses**
The experimental results are too weak.

1. Self-supervised learning aims to transfer the learned representations or whole network parameters into various downstream tasks. However, I do not see any transfer learning experiments in this paper. Could you provide more transfer learning experiments, for example, linear evaluation and fine-tuning in fine-grained classification tasks, semi-supervised learning, and object detection/segmentation?

2. The improvements of this method are very marginal. From Table 1 and Table 2, ACA-Full only surpasses the second-best performance 0.5% in most cases, which needs to be more convincing.

3. The comparison methods in Table 1 and Table 2 are outdated. I highly recommend the author compare ACA-Full with the latest contrastive learning methods. For example, [1][2][3][4][5][6][7]. Moreover, the convergence rate and final accuracy highly depend on the methods. For ImageNet experiments, the authors should at least train the model for 200 epochs (or even longer) to make sure all methods are fully converged.

[1] With a Little Help from My Friends: Nearest-Neighbor Contrastive Learning of Visual Representations
[2] Solving Inefficiency of Self-supervised Representation Learning
[3] Unsupervised Learning of Visual Features by Contrasting Cluster Assignments
[4] Mean Shift for Self-Supervised Learning
[5] Ressl: relational self-supervised learning with weak augmentation
[6] Barlow Twins: Self-Supervised Learning via Redundancy Reduction
[7] AdCo: Adversarial Contrast for Efficient Learning of Unsupervised Representations From Self-Trained Negative Adversaries



**Summary Of The Paper:**

This paper proposes Augmentation Component Analysis (ACA), which employs the idea of PCA to perform dimension reduction on augmentation features. ACA reformulates the steps of extracting principal components of the augmentation features with a contrastive-like loss. With the learned principal components, another on-the-fly loss embeds samples effectively. ACA learns operable low-dimensional embeddings theoretically preserving the augmentation distribution distances.

**Summary Of The Review:**

See Strength and Weaknesses above

---

> ### Author Response · Authors · 2022-11-15
> **Reply to Reviewer fbkG**
>
> We thank you very much for your precious suggestions. And we appreciate your acknowledgment of the novelty of this paper. We have added transfer experiments to enrich the content. We provide transfer tasks with linear evaluation in Table 3, Section 5.4. Object detection results are in Table 5, Appendix D. ACA can achieve competitive results even with shorter epochs. We also provide the performance curve throughout the training in Figure 3, showing that ACA surpasses SimCLR by a considerable margin.
>
> However, we really think the contribution of this work should be judged by the performance. We aim to present a new idea of self-supervised learning by modeling the similarity via augmentation overlap. It is motivated by the observation that semantically similar create similar views by augmentation. Based on this, we view the augmentation distribution as a special description of a sample. And we propose ACA to efficiently perform dimension reduction on this augmentation feature without explicit computation of its exact values. We think the motivation, idea, and method are more valuable than the results.
>
> We conduct the experiment with the following purposes (1) **the motivation is correct**. In the pilot study (Now moved to appendix B), we show **the sample pairs from the same class have smaller augmentation distances than pairs from different classes** (Figure 5 left). Also, the **augmentation feature itself is a discriminative representation** (Figure 5 right, the AF line). In Appendix C, we show that **how views of samples overlap influence the quality of learned embeddings**. (2) **the method is effective**. In the pilot study and experiments on real-world datasets, our ACA method achieves competitive and better results against traditional contrastive learning methods without introducing extra mechanisms like mining nearest neighbors, mixup, momentum encoder, or memory banks. It should be enough to show the effectiveness of the loss.  (3) **The resemblance to contrastive learning in loss form and performance may help explain the success of contrastive learning**.
>
> We think the demonstration of the above is more important than the performance. We have tried our best to add the required experiments, but we can not finish all of them due to restrictions on our hardware. We really hope you can change the decision since the novelty is acknowledged by all the reviewers.
>
> ---
>
> **Q1**: The code and pre-trained models are desired. After reading this paper, it is not easy for me to reproduce the results.
>
> **A1**: Code is uploaded. 100-epoch model is [here](https://drive.google.com/file/d/1phucRxe3188sosLIB5o3CprDRpMRePO0/view). ACA is straightforward to implement, as it is shown in Appendix A. The ACA-PC loss is just three more lines of code than SimCLR loss. Two lines of code can implement the Projection loss.
>
> >     # SimCLR Loss
>     loss = self.cross_entropy(logits, labels)
>
> >     # ACA-PC loss
>     pos_mask = F.one_hot(labels, num_classes=logits.size(-1)).bool()
>     pos = torch.mean(logits[pos_mask])
>     neg = torch.mean(logits[~pos_mask] ** 2)
>     loss = -2 * pos + self.K * neg
>
> >     # projection loss
>     target = 0.5 * (z0 + z1)
>     return F.mse_loss(z, target.detach())
>
> ---
>
> **Q2**:  Self-supervised learning aims to transfer the learned representations or whole network parameters into various downstream tasks. ..., and object detection/segmentation?
>
> **A2**: We provide transfer tasks with linear evaluation in Table 3, Section 5.4. Object detection results in Table 5, Appendix D. ACA can achieve competitive results with shorter epochs.
>
> ---
>
> **Q3**: The improvements of this method are very marginal... needs to be more convincing.
>
> **A3**: Our method does not introduce extra mechanisms and only differs from SimCLR by the loss. We add an extra experiment in Section 5.3. Figure 3 shows that with the projection loss ACA improves the performance throughout the training. We admit that compared to other methods, the improvement is marginal. But we think the idea and the method are more valuable than its performance. Describing data by how it is augmented can be a novel way to build self-supervised learning tasks.
>
> ---
>
> **Q4**: The comparison methods in Table 1 and Table 2 are outdated. ....  for 200 epochs (or even longer) to make sure all methods are fully converged.
>
> **A4**. We added the new methods to the related work section. Since this paper introduces a new idea of self-supervised learning, we think it should be compared to the basic method of the old idea (SimCLR) to see its effectiveness. ACA differs from SimCLR only in loss. The results have shown that our ACA loss is more effective than SimCLR. We apologize that we are not able to finish ImageNet training by now. Training on ImageNet is too costly for our device. We need one week to train a ResNet-50 model for 100 epochs. Currently, we can only provide the result of 100 epochs. But we will make up for the results of the following epochs.

---

> ### Author Response · Authors · 2022-11-17
> **Any other questions?**
>
> We have tried our best to enrich the experiment part and show the effectiveness of ACA. Are there any other questions about the contributions of this paper?

---

### Official Review · Reviewer_LVcG · 2022-10-28

**Confidence:** 3
**Correctness:** 3
**Technical Novelty And Significance:** 3
**Empirical Novelty And Significance:** 3
**Recommendation:** 6

**Clarity, Quality, Novelty And Reproducibility:**

The novelty of this paper can be sufficient and it can be easy to reimplement the proposed ACA.

**Strength And Weaknesses:**

Strengths:
+	The perspective of augmentation overlaps is interesting for contrastive learning. Especially for those samples from the same category, the discovery of augmentation overlaps is much useful for representation learning in an unsupervised way.
+	The loss functions proposed in this paper are verified their effectiveness for self-supervised representation learning. Especially, a projection loss is designed for compacting the representation from the same samples, which shares similar sprit with the prototype learning in supervised learning. The difference lies in the projection loss is mainly for the augmentation distribution of a single sample.

Weaknesses:
-	In this exploration of augmentation overlaps, the similarity between different samples don’t seem to be explained very well in this paper. And, there also seems to have no corresponding loss function or theoretic derivation can prove the model could pull the semantically similar samples close, as showed in the left of Figure 1.
-	The analysis of dimension reduction with the idea of PCA is not suitable to derive the similarity constraint in Eq. (4). Besides, the ACA-PC loss is similar to the function appeared in [1].
-	Performance gain of ACA-PC can be very marginal to prove the effectiveness of the proposed ACA.
-	There exist many sentences in this paper that are hard to digest. E.g., “we claim that it is the similarity between the augmentation distributions of samples, …, that reveals the sample-wise similarity better” (page 1), “It seems that we can obtain the solution directly without further learning, however, not only the elements in the augmentation feature are hard to calculate, but also such high-dimensional target embeddings are impractical to use” (page 1),, “In addition, the resemblance between the objectives of ACA and traditional contrastive loss may explain why the latter can learn semantic-related embeddings” (page 2).

[1] HaoChen J Z, Wei C, Gaidon A, et al. Provable guarantees for self-supervised deep learning with spectral contrastive loss[J]. Advances in Neural Information Processing Systems, 2021, 34: 5000-5011.


**Summary Of The Paper:**

This paper empirically claims that the overlap among the augmentation distributions from a similar category is much higher than those from dissimilar categories. Based on this discovery, this work establishes the semantic relationship between samples in an unsupervised manner based on the similarity of augmentation distributions. Technically, the paper proposes a new self-supervised loss function via maximizing the similarity of samples from the same augmentation distribution and making those from different augmentation distribution as orthogonal as possible for further enhancing the representation learning. Theoretically, this work is intended to explain why contrastive learning can effectively work with its proposed Augmentation Component Analysis.

**Summary Of The Review:**

The paper need further clarify the mentioned items before it is accepted for publishing.

---

> ### Author Response · Authors · 2022-11-15
> **Reply to Reviewer LVcG**
>
> We thank you very much for your precious suggestions and the interesting understanding from the aspect of prototype learning. About the mentioned questions, our answers are as follows:
>
> ---
>
> **Q1**: In this exploration of augmentation overlaps, the similarity between different samples don't seem to be explained very well in this paper. And, there also seems to have no corresponding loss function or theoretic derivation can prove the model could pull the semantically similar samples close, as showed in the left of Figure 1.
>
> **A1**: During the self-supervised learning phase, there is no concept of "semantics”. The meaning of semantics depends on what kind of downstream tasks it applies to. However, on the common downstream tasks, e.g., linear evaluation and object detection, semantically similar samples tend to create similar augmentation results. Therefore, this paper is based on the assumption that such a correlation exists. Thm 4.3 has proved that the embeddings learned by ACA-Full are almost isometric for weighted augmentation distances. This is a weighted squared Euclidean distance between the augmentation distribution. If the correlation assumption holds, semantically similar samples are close in weighted augmentation distances and close in embedding space according to thm 4.3. Section 5 uses a pilot study to show the correctness of weighted augmentation distance. Semantically similar samples have small distances compared to semantically different ones. In Appendix C, we illustrate the histogram of augmentation distances by applying different augmentations and show that the augmentation overlaps play an important role in learning high-quality embeddings.
>
> ---
>
> **Q2**: The analysis of dimension reduction with the idea of PCA is not suitable to derive the similarity constraint in Eq. (4). Besides, the ACA-PC loss is similar to the function appeared in [1].
>
> **A2**: Sorry, I do not get it. Do you mean the similarity constraint in Eq.(1)? Eq.(1) requires the distance measure of learned embedding to be the isometry of augmentation distance. By Thm 4.3, ACA learns a kind of almost isometry.
>
> Yes, ACA-PC is similar to the spectral loss in [1]. This is attributed to the similarity in traditional spectrum-based methods. These methods perform eigendecomposition to the feature or similarity matrix and pick the top eigenvalues/eigenvectors. However, the motivation of our method is different from [1]. [1] learns embeddings for the augmented data by constructing a similarity matrix according to how they are augmented. In contrast, we aim to preserve the similarity of augmentation distributions. Besides, ACA-PC is only a part of our method. We show that ACA-PC preserves certain similarities by Thm 4.2, which is not revealed in [1]. ACA-PC plus projection loss is the final method of this paper. In this revision, we add an extra experiment to show that projection loss not only speeds up the convergence but also improves performance. It is illustrated in figure 3 in the revised version. Without the projection loss, ACA-PC can only achieve similar results as SimCLR, but with projection loss, there is a large improvement.
>
> ---
>
> **Q3**: Performance gain of ACA-PC can be very marginal to prove the effectiveness of the proposed ACA.
>
> **A3**: The comparison in figure 3 can show its effectiveness. This paper does not take performance as the main contribution. We think this paper's value is presenting a new way of self-supervised learning by utilizing augmentation. We view the augmentation distribution as a description of the data, and learn embeddings to preserve relationships provided by such description. To the best of our knowledge, we are the first to take a trial in this way. Also, there are theoretical results to show the properties of learned embeddings: they are almost isometry for the augmentation distribution. This new idea may not currently lead to SOTA results since many methods are designed to polish traditional methods by many other mechanisms like mining nearest neighbors. However, we think the idea of this method should outweigh its produced results.
>
> ---
>
> **Q4**: There exist many sentences in this paper that are hard to digest.
>
> **A4**: Thanks for pointing them out. We polished them in this revision.

---

> > ### Comment · Reviewer_LVcG · 2022-12-02
> > **Comments are addressed**
> >
> > All my concerns are addressed.

---

> > > ### Author Response · Authors · 2022-12-03
> > > **Glad to hear that**
> > >
> > > We are glad that we have addressed your concerns.

---

### Author Response · Authors · 2022-11-15
**General Response**

We thank all the reviewers for their precious suggestions. We are happy that all the reviewers acknowledge the "novelty" of this paper (reviewer LVcG, fbkG, 1C2x, gr8r). The method is "new", "interesting" (reviewer fbkG, 1C2x, gr8r), "solid" and "worth to be introduced to the community" (reviewer 1C2x). The theory is "promising"(reviewer fbkG),  "sound and complete"(reviewer gr8r). We appreciate that all the reviewers have given positive recommendations by far (reviewer LVcG, fbkG, 1C2x, gr8r).

This paper aims to propose a new way of self-supervised learning by utilizing augmentation, and we do not take the performance of this method as the main contribution. We summarize our method here:

**Motivation**: Semantically similar samples usually create similar views. So we can model the similarity of samples by the overlap degree of their augmented views.

**Idea**: Taking the augmentation distribution as the feature of a sample, the similarity measured by this feature reflects the overlap. We call it the augmentation feature. The augmentation feature is very high-dimensional and impractical to use. So we can formulate the embedding learning task as the dimension reduction on the augmentation feature.

**Method**: We propose ACA which simulates the process of PCA with a contrastive-like loss to learn principal components and an on-the-fly projection loss to embed data. ACA can be optimized efficiently by Monte-Carlo sampling without direct calculation of the augmentation feature. Theoretically, the distance of the learned embeddings is an almost isometry of the augmentation distances.

We conduct the experiments not to show it is a SOTA method, but with the following purposes (1) **The motivation is correct**. In the pilot study (now moved to appendix B), we show **the sample pairs from the same class have smaller augmentation distances than pairs from different classes** (Figure 5 left). Also, the **augmentation feature itself is a discriminative representation** (Figure 5 right, the AF line). In Appendix C, we show that **how views of samples overlap influence the quality of learned embeddings**. (2) **The method is effective**. In the pilot study and experiments on real-world datasets, our ACA method achieves competitive and better results against traditional contrastive learning methods without introducing extra mechanisms like mining nearest neighbors, mixup, momentum encoder, or memory banks. It should be enough to show the effectiveness of the loss. (3) **The resemblance to contrastive learning in loss form and performance may help explain the success of contrastive learning**.

Considering the positive aspects of this work, we hope our paper can be accepted and informed to the community.

---

### Decision · Program_Chairs · 2023-01-20

**Decision:**

Accept: poster

**Justification For Why Not Higher Score:**

I'm not against spotlight. However, considering the review scores and the the quality of the submission, I think the poster is appropriate.

**Justification For Why Not Lower Score:**

all reviewers are positive and I also agree with reviewers assessments.

**Metareview: Summary, Strengths And Weaknesses:**

In this paper, the authors propose a novel self-supervised learning method that makes the distance in the embedding space of samples proportional to the distance of the augmentation distribution, starting from the assumption that semantically similar samples have similar sets of views/augmentations. All four reviewers agreed that the idea is interesting and novel and the proposed method is theoretically sound and complete. In the original review, there were concerns about presentation clarity and limited experimentations, but through the rebuttal period, the authors successfully addressed these concerns, and there was no disagreement in accepting the paper in the discussion.

**Note From Pc:**

if the above contains the word "oral" or "spotlight" please see: "oral" presentation means -> notable-top-5% and "spotlight" means -> notable-top-25%. As stated in our emails, we are disassociating presentation type from AC recommendations